# EVOCURR: SELF-EVOLVING CURRICULUM WITH BEHAVIOR CODE GENERATION FOR COMPLEX DECISION-MAKING

## ABSTRACT

While large language models (LLMs) demonstrate remarkable capabilities across diverse domains, they fail catastrophically on high-complexity tasks requiring long-horizon reasoning and multi-step coordination. To address this problem, we present EvoCurr, a self-evolving curriculum learning framework that enables LLMs to solve complex decision-making problems through cooperative multi-agent learning. The core of EvoCurr is a multi-agent cooperative system where a Designer agent generates adaptive task sequences and a Solver agent produces executable solutions through coordinated interaction. Both agents share identical rewards based on task performance and proximity to the target task, creating a fully cooperative framework that naturally aligns their objectives for progressive skill acquisition. A critical innovation is the accepted-floor constraint that prevents difficulty regression below previously solved levels, ensuring monotonic skill advancement while preventing catastrophic forgetting. The framework enforces feasibility through a validation gate and supports both open-loop code generation and closed-loop policy learning paradigms. We evaluate EvoCurr on two complementary domains: StarCraft II micro-management and Overcooked coordination tasks. On StarCraft II micro-management, where the Solver generates Python behavior-tree scripts for complex tactical scenarios, EvoCurr achieves average combat winning rates above 90% while state-of-the-art models achieve less than 50% when directly attempting these scenarios. On Overcooked coordination tasks, where the Solver uses multi-agent reinforcement learning to train cooperative policies, EvoCurr achieves 20% higher task completion rates (measured by dish orders delivered) compared to direct training. Our results demonstrate that EvoCurr provides a principled, domain-agnostic approach for extending LLM capabilities to complex decision-making tasks previously beyond their reach.

## 1 INTRODUCTION

Large language models (LLMs) have revolutionized automated problem-solving, from synthesizing formal proofs to generating executable Python programs Brown et al. (2020); OpenAI (2023); Bubeck et al. (2023); Chen et al. (2021); Li et al. (2022). Yet when faced with truly complex decision-making tasks—those requiring long-horizon planning, multi-step coordination, and adaptive strategies—even the most advanced models struggle dramatically. Consider StarCraft II micro-management: controlling dozens of military units with diverse abilities against sophisticated opponents. When asked to generate control code for such scenarios directly, GPT-5, Claude-4, DeepSeek-3.1, and Gemini-2.5 achieve less than 50% win rates, despite these tasks being well within human capability Zelikman et al. (2022). This performance gap reveals a fundamental challenge: while LLMs possess vast knowledge, they cannot effectively marshal this knowledge for complex, multi-step decision problems.

The core issue is complexity scaling. Simple tasks succeed reliably, but compound tasks—such as coordinating 20 Marines, 8 Ghosts with cloaking, and 4 Medivacs for healing while engaging enemy Protoss forces—overwhelm even the most capable models. The failure is not due to lack of knowledge; these models understand unit capabilities, tactical concepts, and programming interfaces. Rather, they cannot synthesize this knowledge into working solutions when the problem

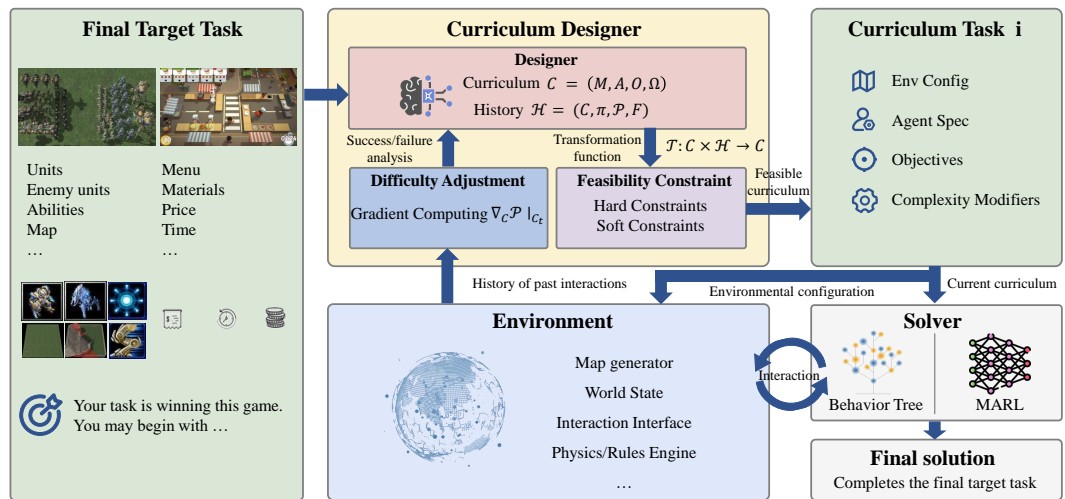

Figure 1: Brief introduction for EvoCurr. Showing the framework of EvoCurr that the designer gains advantage by generating feasible yet demanding tasks that progress toward the objective, whereas the Solver profits from mastering increasingly difficult challenges and ultimately achieving the target goal.

space becomes too large. Two control paradigms illustrate this challenge concretely. In *open-loop* control, one compiles an interpretable program (e.g., a behavior tree) and runs it without adapting to new observations; this eases debugging but is sensitive to missing cases. In *closed-loop* control, one learns a reactive policy mapping observations to actions (typically via reinforcement learning); this improves robustness but sacrifices transparency. A mechanism that can *at inference time* progress from easy to hard tasks in both paradigms—without retraining the base model—would substantially increase the practical utility of LLM-based decision making.

Humans don't learn complex skills by jumping directly to the hardest version. A chess player starts with basic piece movements before attempting complex strategies. This observation suggests a natural solution: can we enable LLMs to solve complex problems by automatically discovering and following a learning curriculum? Curriculum learning has proven effective for graduated complexity Bengio et al. (2009); Graves et al. (2017); Narvekar et al. (2020); Narvekar & Stone (2018), but three obstacles limit its use for LLM inference. First, curricula typically require domain expertise and manual task design, which is expensive and brittle. Second, most approaches optimize training-time schedules and offer little guidance for inference-time problem solving with pretrained models. Third, existing practices lack a simple, verifiable rule for when and how to escalate difficulty while avoiding catastrophic forgetting once a skill threshold has been reached.

We propose *EvoCurr*, a self-evolving curriculum framework that enables LLMs to solve complex decision-making problems they cannot handle directly. The key insight is that LLMs themselves can design appropriate curricula—they understand what makes tasks easier or harder and can propose suitable stepping stones toward a final goal. EvoCurr instantiates this as a cooperative two-agent system. A *Designer* analyzes current capabilities and proposes the next task by adjusting controllable factors (e.g., in StarCraft II: unit composition, abilities, and map; in Overcooked: layout, recipes, and timing). A *Solver* produces an executable solution, evaluates it on the proposed task, and returns the outcome.

Two simple rules make this loop progress reliably without manual intervention. First, the *accepted-floor* rule remembers the most recently mastered task and forbids future proposals from going easier than that point—once a skill is demonstrated, the system maintains this skill floor, preventing catastrophic forgetting. Second, a *feasibility gate* discards ill-formed proposals early by checking basic validity: the task compiles (syntax), its logic allows the goal to be attempted (e.g., reachable waypoints), and it can be run to produce a measurable outcome (runtime). With just a single acceptance threshold defining "mastery" (e.g., winning rate above 90%), these rules let EvoCurr autonomously navigate the frontier of learned capabilities without hand-crafted schedules or domain-specific dif-

ficulty metrics. Crucially, the same framework applies to both control paradigms: the Solver either generates executable behavior-tree code (*open-loop code-as-policy*) or trains a reactive policy for a fixed budget (*closed-loop*).

We validate EvoCurr on two challenging domains that have resisted direct LLM approaches. In StarCraft II micro-management across twelve complex combat scenarios, EvoCurr progressively achieves winning rates exceeding 90% by generating sophisticated behavior-tree scripts, while direct one-shot generation with the same models achieves less than 50%. The evolution typically requires 4-6 intermediate tasks, automatically discovered by the system, to bridge from simple unit control to complex multi-unit coordination with advanced abilities. In Overcooked, a challenging multi-agent coordination benchmark, EvoCurr achieves 20% higher task completion rates (measured by successfully delivered orders) compared with direct training under matched total budgets. The framework discovers curricula that first master basic movement and item handling, then progress to timing-critical coordination in confined spaces. These results demonstrate that an inference-time curriculum—implemented by simple "do not go backwards" and "only propose valid tasks" rules—can reliably unlock LLM capabilities for complex decision-making previously beyond their reach.

Summarizing, our contributions are:

1. **Inference-time curriculum mechanism.** A self-evolving framework that advances task difficulty using only an acceptance threshold, an *accepted-floor* rule preventing skill regression, and a *feasibility gate* filtering invalid proposals—eliminating manual curriculum design and domain-specific difficulty metrics.

2. **Practical Designer–Solver procedure.** The Designer diagnoses capability bottlenecks from historical outcomes and proposes targeted task adjustments; the Solver produces executable artifacts and measured performance, forming an autonomous improvement loop that works across both open-loop code generation and closed-loop policy learning paradigms.

3. **Empirical evidence across domains.** On StarCraft II micro-management, EvoCurr progressively attains winning rates $\geq 90\%$ where direct generation fails; on Overcooked, with matched budgets, EvoCurr achieves 20% higher completion rates, demonstrating that inference-time curriculum evolution can extend LLM capabilities to complex tasks previously beyond their reach.

## 2 RELATED WORK

**Curriculum Learning.** Bengio et al. (Bengio et al., 2009) formalized curriculum learning, demonstrating that training on examples organized from easy to hard improves generalization and convergence compared to random data shuffling. This paradigm has achieved success across computer vision, NLP, and reinforcement learning (Soviany et al., 2022; Wang et al., 2021b). Kumar et al. (Kumar et al., 2010) introduced self-paced learning (SPL) where models automatically determine learning pace based on sample difficulty, eliminating predefined curricula. Jiang et al. (Jiang et al., 2015) extended SPL with diversity constraints to prevent premature convergence. In reinforcement learning, Narvekar et al. (Narvekar et al., 2020) provided a comprehensive curriculum framework, while Klink et al. (Klink et al., 2020) interpreted curriculum generation as an inference problem. Recent advances include Teacher-Student Curriculum Learning (Matiisen et al., 2019) with teacher networks generating student tasks, and Prioritized Level Replay (Jiang et al., 2021) sampling training levels based on learning potential. However, these approaches primarily focus on training phase optimization and require either manual curriculum design or domain-specific difficulty metrics, leaving a gap for inference-time adaptive curriculum generation.

**Environment Generation.** Procedural content generation has evolved from rule-based methods to learning-based approaches (Liu et al., 2021a). POET (Wang et al., 2019) co-evolves agents and environments through population-based training, while PAIRED (Dennis et al., 2020) uses adversarial training to generate challenging yet solvable environments. EnvGen (Zhai et al., 2024) leverages LLMs to adaptively create training environments for RL agents, using world knowledge to generate environment configurations based on task descriptions. Samvelyan et al. (Samvelyan et al., 2023) introduced Rainbow Teaming for diverse adversarial scenarios. Recent work explores evolution

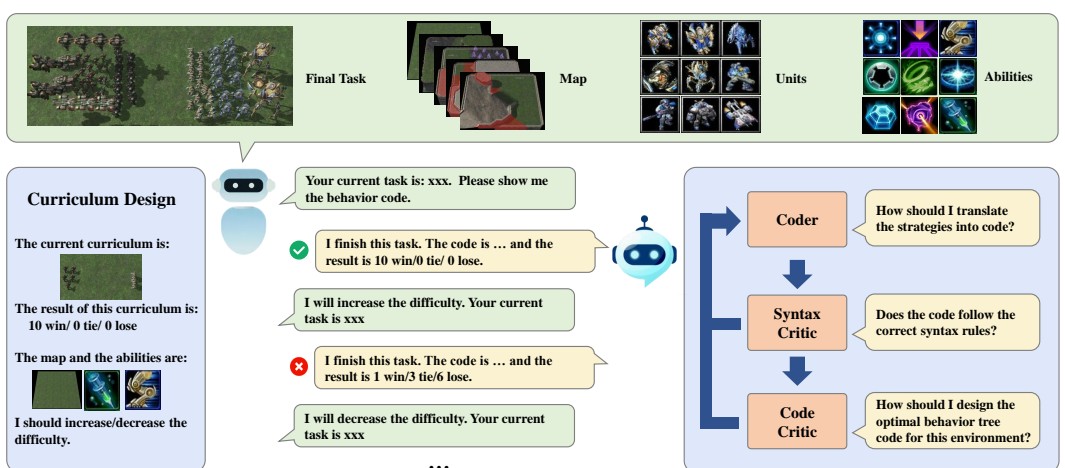

Figure 2: EvoCurr overview. A curriculum designer proposes the next curriculum $C_{t+1}$; a solver produces an executable policy $\pi_{t+1}$ and evaluates it; the outcome feeds back to the designer. The loop starts from a simplified version of the final target $T_f$ and proceeds until $T_f$ is solved.

strategies for environment generation (Liu et al., 2024) and evolved curricula that transfer across different learners (Parker-Holder et al., 2022). These methods generate static training data or environments before agent training, rather than dynamically adapting during inference based on solver capabilities.

**Code Generation in StarCraft II.** StarCraft II has become a standard benchmark for complex decision-making research following DeepMind's PySC2 (Vinyals et al., 2017). AlphaStar (Vinyals et al., 2019) achieved Grandmaster level through large-scale reinforcement learning, with subsequent work exploring efficient strategies (Liu et al., 2021c;b), offline learning (Mathieu et al., 2021), and federated frameworks (Han et al., 2020; Wang et al., 2021a). Recent integration of language models includes TextStarCraft II (Ma et al., 2024; 2025a;b; Li et al., 2025) and behavior tree approaches (Deng et al., 2025; 2024). Beyond StarCraft II, collaborative environments like Overcooked have emerged as benchmarks for multi-agent coordination (Carroll et al., 2020), with recent work providing comprehensive evaluation toolkits for zero-shot coordination (Wang et al., 2024). For code generation, Liang et al. (Liang et al., 2023) proposed Code as Policies for robot control, while behavior tree synthesis work (Colledanchise & Ögren, 2018; Lykov & Tsetserukou, 2023) demonstrated that LLMs can produce structurally correct trees. These methods successfully generate executable policies but operate on fixed tasks without adaptive difficulty progression. Building upon these foundations, we propose EvoCurr, a framework that enables autonomous curriculum evolution for complex decision-making scenarios through self-adaptive task generation and progressive skill acquisition.

## 3 METHOD

This section presents *EvoCurr*, a framework that enables LLMs to solve complex decision-making tasks through self-evolving curricula. We employ a two-agent cooperative framework (Section 3.1), design a curriculum generation mechanism with feasibility constraints (Section 3.2), and describe the code-as-policy realization for behavior tree synthesis (Section 3.3).

### 3.1 TWO-AGENT COOPERATIVE FRAMEWORK

EvoCurr employs two cooperating agents: a *Designer* that generates curricula and a *Solver* that produces policies. These agents share identical rewards, creating a fully cooperative system where success requires coordinated action across different decision spaces.

Let $\mathcal{C}$ denote the task space containing target $T_f \in \mathcal{C}$, and $\Pi$ the policy space. Each task $C \in \mathcal{C}$ has difficulty $d(C) \in \mathbb{R}_+$ measuring complexity through unit count and ability diversity. The distance

$\Delta(C, T_f)$ quantifies the configuration gap to the final target. The performance function $\mathcal{P} : \Pi \times \mathcal{C} \rightarrow [0, 1]$ evaluates policy $\pi \in \Pi$ on task $C$, typically as win rate over multiple rollouts. The history $\mathcal{H}_t = \{(C_i, \pi_i, \mathcal{P}_i, \text{Accept}_i)\}_{i=1}^t$ records past curricula, policies, performances $\mathcal{P}_i = \mathcal{P}(\pi_i | C_i)$, and acceptance status $\text{Accept}_i = \mathbb{1}[\mathcal{P}_i \geq \tau]$ where $\tau \in (0, 1)$ is the acceptance threshold.

At round $t$, the Designer generates a new curriculum through LLM-based transformation:

$$C_{t+1} = \mathcal{T}(C_t, \mathcal{H}_t, T_f, \text{Accept}_t) \tag{1}$$

Curriculum generation follows the **accepted-floor constraint**. Let $C_{t^*}$ denote the most recently accepted task. Then:

$$\begin{cases} d(C_{t+1}) > d(C_t) \text{ and } \Delta(C_{t+1}, T_f) < \Delta(C_t, T_f) & \text{if Accept}_t = 1 \\ d(C_{t^*}) < d(C_{t+1}) < d(C_t) & \text{if Accept}_t = 0 \end{cases} \tag{2}$$

This ensures monotonic skill acquisition—the system never regresses below previously mastered difficulty levels.

The Solver generates policies via LLM-based code synthesis or neural network training:

$$\pi_{t+1} = \text{Solver}(C_{t+1}, \mathcal{H}_t) \tag{3}$$

$$\text{Accept}_{t+1} = \mathbb{1}[\mathcal{P}(\pi_{t+1} | C_{t+1}) \geq \tau] \tag{4}$$

Both agents optimize toward high performance on progressively harder tasks approaching $T_f$, with shared incentives ensuring the Designer proposes solvable challenges while the Solver develops increasingly sophisticated policies.

### 3.2 Curriculum Generation and Feasibility Constraints

A task $C = (M, A, G)$ consists of map configuration $M$, agent specifications $A = \{a_i\}_{i=1}^n$ where $a_i = (\text{type}_i, \text{count}_i, \text{abilities}_i)$, and goal $G$. The Designer uses history $\mathcal{H}_t$ to identify capability bottlenecks: coordination failures lead to reduced agent count while maintaining tactical structure; timing issues trigger ability simplification before count adjustment.

For example: Task 2 succeeds with Marine×10, Medivac×2 (90% win rate); Task 3 fails with Marine×15, Ghost×4, Tank×3 (40%); Task 4 adjusts to Marine×12, Ghost×2 (90%), ensuring $d(\text{Task 2}) < d(\text{Task 4}) < d(\text{Task 3})$ per the accepted-floor constraint.

A feasibility gate $g_{\text{feas}}$ validates curricula through syntax checking (code compilation), logic verification (path reachability), and runtime validation (execution success).

### 3.3 Code-as-Policy: Behavior Tree Synthesis

The Solver adapts its policy generation based on the control paradigm required by the task domain.

For open-loop control requiring interpretable policies, the Solver generates executable behavior tree code through three stages: (1) strategic planning extracts high-level objectives $S$ from $C_{t+1}$; (2) code synthesis translates $S$ into structured behavior trees; (3) compilation produces the final policy $\pi_{t+1}$. On failure ($\mathcal{P} < \tau$), the system adjusts decision thresholds and action priorities based on performance feedback.

For closed-loop control requiring continuous adaptation, the Solver trains neural policies via RL algorithms, with $\pi_{t+1}$ representing network parameters optimized in environment $C_{t+1}$. Training continues for a fixed timestep budget before evaluation. Both paradigms share the same cooperative dynamics, accepted-floor constraints, and performance evaluation, enabling EvoCurr to handle diverse decision-making challenges within a unified framework.

## 4 Experiments

We evaluate EvoCurr in two complementary domains that demonstrate its versatility across different control paradigms. In StarCraft II micro-management, we transform the traditionally closed-loop problem into open-loop control: the Solver generates complete behavior tree scripts upfront

that execute without real-time adaptation, departing from typical RL approaches (Samvelyan et al., 2019; Vinyals et al., 2017) that react at each timestep. This code-as-policy approach tests whether LLMs can tackle reactive domains through strategic pre-planning while producing interpretable solutions. Conversely, in Overcooked (Carroll et al., 2020), the Solver trains MARL policies that continuously adapt to observations, maintaining the conventional closed-loop paradigm. Despite these fundamentally different policy realizations—pre-compiled behavior trees versus learned neural networks—both operate under the same EvoCurr framework with the feasibility gate $g_{\text{feas}}$ and accepted-floor constraint ensuring monotonic progression. Implementation details are in Appendices C and A.1.

| AGENTS (Terran): | | | ENEMIES (Protoss): | | |
|---|---|---|---|---|---|
| **Unit Type** | **Quantity** | **Technology** | **Unit Type** | **Quantity** | **Technology** |
| Marine | 20 | Stimpack | Zealot | 15 | Charge |
| Marauder | 12 | Stimpack | Stalker | 14 | BlinkTech |
| Medivac | 4 | Heal | Sentry | 10 | ForceField |
| Ghost | 8 | PersonalCloaking | HighTemplar | 8 | PsiStormTech |
| SiegeTank | 6 | SiegeTech | Colossus | 4 | ExtendedThermalLance |
| VikingFighter | 8 | AssaultMode | Tempest | 5 | GroundAttack |
| Cyclone | 7 | LockOn | Disruptor | 4 | PurificationNova |
| WidowMine | 7 | Burrow | Carrier | 4 | InterceptorLaunch |
| Raven | 3 | HunterSeeker | | | |
| Liberator | 2 | DefenderMode | | | |

Table 1: Final Terran vs Protoss Task Specification

| AGENTS (Terran): | | | ENEMIES (Zerg): | | |
|---|---|---|---|---|---|
| **Unit Type** | **Quantity** | **Technology** | **Unit Type** | **Quantity** | **Technology** |
| Marine | 20 | Stimpack | Zergling | 60 | ZerglingMovementSpeed |
| Marauder | 12 | Stimpack | Baneling | 24 | CentrificalHooks |
| Medivac | 4 | Heal | Roach | 15 | GlialReconstitution |
| Ghost | 8 | PersonalCloaking | Hydralisk | 10 | HydraliskSpeed |
| SiegeTank | 6 | SiegeTech | Lurker | 6 | Burrow |
| VikingFighter | 8 | AssaultMode | Corruptor | 10 | FlyerWeaponsLevel1 |
| Cyclone | 7 | LockOn | Infestor | 3 | EnergyUpgrade |
| WidowMine | 7 | Burrow | Viper | 4 | FlyerArmorsLevel1 |
| Raven | 3 | HunterSeeker | Overseer | 3 | FlyerArmorsLevel1 |
| Liberator | 2 | DefenderMode | Queen | 4 | MissileWeaponsLevel1 |
| | | | Broodlord | 4 | FlyerWeaponsLevel1 |

Table 2: Final Terran vs Zerg Task Specification

### 4.1 STARCRAFT II MICRO-MANAGEMENT

**Experiment Setup** The Solver generates `python-sc2` behavior trees that act at the unit-action level and is evaluated in an open-loop manner. We test on five newly designed micro maps against two opponent races (Terran vs Protoss and Terran vs Zerg). Each curriculum specifies unit sets, technologies, and spawn regions on a selected map; *compile-and-run* serves as a hard feasibility gate in line with $g_{\text{feas}}$. The final target $T_f$ for the canonical Terran–Protoss and Terran–Zerg settings is given in Table 1 and Table 2. For acceptance, we require $\mathcal{P}(\pi|C) \geq \tau = 0.9$, evaluated as win rate over 10 rollouts. The primary baseline, *Direct Code*, attempts to solve $T_f$ in one shot under the same rollout and validation budgets as EvoCurr. Per-curriculum compositions and complete evolution traces are summarized in the appendix.

Because direct long-horizon code generation can be brittle (syntax/API errors) and win rate alone may not capture partial successes, we additionally report a damage-cost-aware combat score $\mathcal{S}_{combat}$ for more nuanced evaluation. This metric evaluates the comparative performance of EvoCurr against closed-source large language model performances (DeepSeek3.1, GPT-5, Claude4, Gemini2.5) on

direct target task implementation., given by:

$$\mathcal{S}_{combat} = 0.5 \cdot \frac{R_{\text{agent\_final}}}{R_{\text{agent\_init}}} + 0.5 \cdot \left(1 - \frac{R_{\text{enemy\_final}}}{R_{\text{enemy\_init}}}\right), \tag{5}$$

where the total combat power for one side is

$$R_{\text{side}} = \sum_i \left(\text{minerals}_i + \alpha \cdot \text{vespene}_i + \beta \cdot \text{build\_time}_i\right) \cdot \frac{\text{hp}_i + \text{shields}_i}{\text{hp\_max}_i + \text{shields\_max}_i}. \tag{6}$$

This metric aggregates resource cost and remaining health/shields; scores range from 0 to 1. A combat score $\mathcal{S}_{combat} > 0.5$ indicates successful annihilation of the majority of enemy forces while preserving our own, providing a finer-grained assessment than binary win/loss especially for failed code executions where the first term becomes 0.

| Task | Win Rate EvoCurr (%) | Win Rate DeepSeek (%) | Score EvoCurr | Score DeepSeek | Task nums |
|---|---|---|---|---|---|
| Bush (TvP) | 100 | 0 | 0.67 | 0.33 | 6 |
| Bush (TvZ) | 100 | 100 | 0.73 | 0.61 | 4 |
| Corridor (TvP) | 100 | 80 | 0.71 | 0.66 | 5 |
| Corridor (TvZ) | 100 | 90 | 0.69 | 0.58 | 5 |
| Main (TvP) | 100 | 100 | 0.72 | 0.69 | 4 |
| Main (TvZ) | 100 | 100 | 0.70 | 0.68 | 5 |
| Ramp (TvP) | 100 | 10 | 0.72 | 0.45 | 4 |
| Ramp (TvZ) | 100 | 50 | 0.74 | 0.45 | 5 |
| Corner (TvP) | 90 | 90 | 0.55 | 0.56 | 5 |
| Corner (TvZ) | 100 | 30 | 0.84 | 0.41 | 5 |
| Flat (TvP) | 100 | 0 | 0.69 | 0.41 | 5 |
| Flat (TvZ) | 100 | 0 | 0.62 | 0.36 | 5 |

Table 3: Combat score $\mathcal{S}_{\text{combat}}$ among *correct* scripts on StarCraft II micro tasks. T denotes *Terran*, P *Protoss*, Z *Zerg*. Each entry is the maximum over 10 evaluations. Task nums denotes the nums of tasks to complete the target task.

**Experimental Results** We evaluate 12 complex micro-management tasks (2 matchups $\times$ 6 maps) in Table 3. EvoCurr achieves most of the highest combat scores (often exceeding 0.7), demonstrating consistent performance regardless of task difficulty. When one-shot code generation proved challenging for all models (scores ¡ 0.5), EvoCurr reliably achieved scores near 0.7. Conversely, on simpler tasks where most one-shot methods scored above 0.5, EvoCurr still delivered robust, high-performing results, though not necessarily the peak score, aligning with its goal of final task accomplishment through progressive curriculum advancement.

## 4.2 OVERCOOKED

| Map | Task | Orders | Agent0 Delivery | Agent1 Delivery | Total Delivery | Sparse Reward |
|---|---|---|---|---|---|---|
| Map 1 | EvoCurr | 5 | 14.74 | 14.36 | 29.10 | 290.9 |
| Map 1 | Direct Training | 5 | 11.93 | 12.18 | 24.11 | 240.5 |
| Map 2 | EvoCurr | 5 | 7.82 | 10.82 | 18.64 | 186.2 |
| Map 2 | Direct Training | 5 | 7.40 | 8.96 | 16.36 | 163.4 |

Table 4: Curriculum progression and performance metrics for overcook maps

**Experiment Setup** We instantiate EvoCurr in a closed-loop regime on Overcooked while keeping the game dynamics consistent with Section 3. The Designer proposes curricula parameterized by layouts, ingredient placements, order timing constraints, and stochasticity. Unlike the StarCraft II setting, the Solver here is a MARL trainer that optimizes decentralized policies under a fixed per-curriculum budget $B = 10^7$ timesteps. The acceptance criterion $\mathcal{P}(\pi|C) \geq \tau$ is defined as completing all required orders, where $\mathcal{P}(\pi|C) = 1$ if all orders are fulfilled and 0 otherwise. The framework also allows completing bonus orders after required ones, contributing to the total delivery count shown in Table 4. Orders define the prescribed objectives for agent operations, where

reward structures are calibrated to provide greater compensation for deliveries that exceed the established order quantities. This isolates the effect of inference-time curriculum evolution from policy realizations.Detailed curriculum specifications and complete performance metrics for both map configurations are provided in Appendix D.

**Experimental Results** As shown in Table 4, EvoCurr outperforms directly applying ET3 under matched total budgets (After testing the budgets that need to be used with EvoCurr, then directly conduct testing using the same budget). On the first task, EvoCurr achieves 29.1 effective deliveries on average vs 24.11 for the baseline; on the second, EvoCurr reaches 18.64 vs 16.36. The Map1 is Coord. Ring with Multi-recip and the Map 2 is Counter Circuit with Multi-reci (Wang et al., 2024). The higher delivery counts for EvoCurr include both required and bonus orders, demonstrating that the progressive curriculum not only ensures completion of primary objectives but also enables more efficient exploration of bonus rewards. Performance drops at curriculum transitions reflect distribution shift: policies specialized to one curriculum must re-explore when difficulty increases, yet prior experience accelerates re-convergence—consistent with the progressive, monotone advancement prescribed by the framework.

## 5 ANALYSIS

**Curriculum Design** We show the effectiveness of the curriculum designing module by providing the sub-tasks generated for solving the final task in Table 5. The sub-tasks are designed based on the accomplishment of the previous curriculum. For StarCraft II tasks, we set the acceptance threshold $\tau = 0.9$, which indicates that the difficulty should increase when the winning rate is above 90% during the evaluation process and in contrast decrease otherwise. In the 12 new tasks, according to the table, the Terran vs Protoss setting on the map of Bush takes the longest curriculum trajectory. The enemies are invisible in the bush, which brings challenges to the agents, so the designer has to decrease the difficulty twice before the solver finally finish the task.

| Final Task | Task 1 | Task 2 | Task 3 | Task 4 | Task 5 | Task 6 | Task 7 |
|---|---|---|---|---|---|---|---|
| Flat (TvP) | 100% | 100% | 70% | 90% | 100% | - | - |
| Flat (TvZ) | 100% | 100% | 100% | 100% | 100% | - | - |
| Bush (TvP) | 100% | 100% | 40% | 90% | 50% | 100% | 100% |
| Bush (TvZ) | 100% | 90% | 60% | 100% | 100% | - | - |
| Corridor (TvP) | 100% | 90% | 100% | 100% | 90% | - | - |
| Corridor (TvZ) | 100% | 90% | 90% | 100% | 100% | - | - |
| Corner (TvP) | 90% | 90% | 90% | 100% | - | - | - |
| Corner (TvZ) | 100% | 100% | 90% | 90% | 100% | - | - |
| Main (TvP) | 100% | 100% | 100% | 100% | - | - | - |
| Main (TvZ) | 100% | 100% | 100% | 100% | - | - | - |
| Ramp (TvP) | 100% | 100% | 90% | 100% | 100% | - | - |
| Ramp (TvZ) | 100% | 100% | 100% | 90% | 100% | - | - |

Table 5: The winning rates of each curricula designed for the 12 complex decision-making scenarios.

Figure 3 also demonstrates curriculum paths on StarCraft II micromanagement tasks solved by open-loop behavior trees and Overcook scenarios solved by MARL algorithms. Given a final task, the designer determines the first class with limited difficulty. The solver starts to finish the task and respond the rollout results to the designer. Then the designer generates new curriculum based on the rollout results. In the Figure, the solver achieves more than 90% winning rates in the curricula which are shown in cyan color. When the solver cannot finish the task, red points, the designer then selects the latest finished task as the basement and generates new curriculum with different map settings. When the solver solves the final task, the tree-based evolution process is terminated and the final behavior tree/black-box policy model are returned as the final solution to the task.

**Behavior Coder Generation** When facing new curriculum with larger unit amount and new unit type, the behavior coder generates new scripts based on the previous script that finish the previous curriculum. The new scripts are refined in two phases. The first phase is the addition of new control

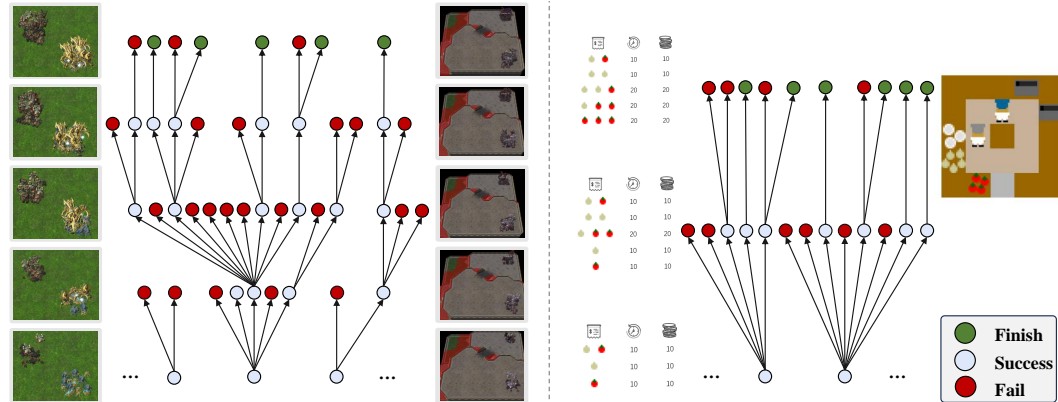

Figure 3: A demonstration of the evolution traces from the initial task to the final tasks in StarCraft II setting and the overcook settings. In the StarCraft II, the curriculum is designed based on the maps, units, and abilities. In the overcook scenarios, the curricula are designed on the recipes, times, and orders.

functions. As shown in Code A.3.2 and Code A.4 in the Appendix, the advanced solution contains more control functions for each unit type, such as control_vikings, control_ghosts, etc. The second phase is the refinement of each existing control function which promotes the coordination among units. For instance, in the early coding stage, the Marine units are responsible for focusing fire on the enemy and rapidly decrease the enemy units. In the latter curriculum where the enemy has the area of effect (aoe) attacking ability, the Marines should firstly split to avoid aoe attacks and then focus fire on the enemy. In such case, the split skill is learned during the evolution and the fire focus skill is reserved and promoted.

**Layered Critic Refinement**    Despite that the LLMs have learnt extensive coding script resouces during the pre-training process. The ability of generating scripts following python_sc2 package depends on the amount of handcrafted python_sc2 scripts from the community, which results in the different code generating ability of different LLMs. Therefore, in the behavior tree generation, we leverage a two-layered critic to improve the quality of behavior tree. The first layer is the sanity check module that is responsible for correcting the potential mistakes such as grammar bugs, API misuse, and exceptions. The second layer of the critic refines the strategy which provides suggestion on the implementation logic of the behavior tree scripts. The two-layer critic module serves as a critical support to the solver for higher success behavior tree generating rates.

## 6  DISCUSSION, FUTURE WORK, AND CONCLUSION

EvoCurr demonstrates that self-evolving curricula enable LLMs to solve complex decision-making tasks at inference time without manual curriculum design. The cooperative Designer-Solver framework, constrained by the accepted-floor rule and feasibility gating, achieves systematic progression toward target tasks across both open-loop behavior tree generation and closed-loop MARL training—reaching 90% win rates in StarCraft II where direct approaches achieve only 50%, and 20% higher task completion in Overcooked. Key limitations include sensitivity to difficulty scaling leading to rejection cycles, LLM context constraints limiting behavior tree complexity, and computational overhead from maintaining historical information $\mathcal{H}_t$. Future directions include hierarchical multi-agent architectures for the Solver to handle complex task decomposition, adaptive difficulty scaling based on acceptance patterns, and hybrid approaches combining behavior tree interpretability with neural policy robustness through distillation. EvoCurr provides a principled, domain-agnostic mechanism for extending LLM capabilities to complex sequential decision-making, offering a practical path toward deployable systems that maintain interpretability while handling tasks previously beyond their reach.

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

## A  STARCRAFT II MICRO MANAGEMENT

### A.1  INTRODUCTION TO STARCRAFT II

StarCraft II is a real-time strategy game developed by Blizzard Entertainment that has become one of the most challenging and strategically complex video games ever created. Released in 2010, the game features three asymmetric factions—Terrans, Protoss, and Zerg—each with distinct units, technologies, and strategic approaches. Players must simultaneously manage multiple interconnected systems: resource collection and allocation, base construction and expansion, technological research and upgrades, unit production and army composition, and real-time tactical combat control. The game demands rapid decision-making under time pressure, long-term strategic planning, adaptation to opponent strategies, and precise micro-management of individual units during combat. Professional matches can involve hundreds of units across multiple battlefronts, requiring players to process vast amounts of information while executing complex multi-layered strategies. The skill ceiling is extraordinarily high, with professional players dedicating years to master the intricate mechanics, build orders, timing attacks, and unit interactions that define high-level play.

The significance of StarCraft II for artificial intelligence research extends far beyond its entertainment value. The game presents a comprehensive testbed for studying complex decision-making under uncertainty, partial information, and real-time constraints challenges that mirror many real-world applications of AI. Unlike traditional board games such as chess or Go, which have perfect information and turn-based mechanics, StarCraft II requires agents to operate in a partially observable environment with continuous action spaces and exponentially large state representations. The game's multi-scale nature demands both macro-level strategic planning spanning tens of minutes and micro-level tactical execution occurring within milliseconds. This dual requirement has driven significant advances in hierarchical reinforcement learning, multi-agent coordination, and long-horizon planning algorithms. Notable breakthroughs include DeepMind's AlphaStar, which achieved Grandmaster level performance and demonstrated that AI systems could master complex strategic reasoning, and subsequent research that has explored everything from curriculum learning and imitation learning to neural architecture search and federated training. The availability of extensive replay datasets, standardized evaluation protocols through environments like PySC2, and the game's inherent interpretability through observable unit actions have made StarCraft II an invaluable platform for developing and benchmarking AI systems capable of human-level strategic reasoning in complex, dynamic environments.

### A.2  STARCRAFT II API AND PYTHON INTERFACES

The technical foundation enabling AI research in StarCraft II rests on Blizzard Entertainment's official StarCraft II Machine Learning API, which provides programmatic access to the game's complete state information and action execution capabilities. This API exposes the game engine through a protocol buffer-based interface that delivers real-time observations including unit positions, resource states, map geometry, and tactical information while accepting high-level commands for unit control, building construction, and technology research. The official **s2client-proto** defines the core communication protocol between external programs and the StarCraft II executable, establishing standardized data structures for observations, actions, and game configuration. This low-level interface handles the complex details of game state serialization, network communication, and command validation, but requires substantial boilerplate code and deep understanding of the underlying protocol specifications to implement effective AI agents.

Building upon this foundation, the research community has developed higher-level abstractions that significantly simplify AI development while preserving the full functionality of the underlying API. **PySC2**, developed by DeepMind, transforms the raw API into a structured reinforcement learning environment that follows standard RL conventions with observation spaces, action spaces, and reward functions. This environment emphasizes feature-layer representations and provides built-in mini-games for curriculum learning, making it particularly suitable for deep reinforcement learning approaches.

Complementing PySC2, the **python-sc2** library offers a more direct and intuitive interface focused on scripted bot development, where complex strategic behaviors can be implemented using straightforward Python code with minimal boilerplate. The python-sc2 library abstracts away protocol

buffer complexities while exposing high-level game objects such as units, abilities, and map structures through clean Python APIs, enabling researchers to focus on strategic logic rather than low-level implementation details. Our EvoCurr framework leverages python-sc2's accessibility and expressiveness to generate behavior tree scripts that can be easily interpreted, debugged, and modified, making it an ideal choice for our curriculum-based approach to complex tactical reasoning.

### A.3 GENERATED BEHAVIOR TREE CODE EXAMPLES

This section presents complete examples of behavior tree implementations generated by the EvoCurr framework at different curriculum stages, demonstrating the evolution of tactical complexity.

### A.3.1 EARLY STAGE: BASIC MARINE MICRO-MANAGEMENT

```python
from sc2 import maps
from sc2.bot_ai import BotAI
from sc2.data import Race, Difficulty
from sc2.ids.ability_id import AbilityId
from sc2.ids.unit_typeid import UnitTypeId
from sc2.main import run_game
from sc2.player import Bot, Computer

class BattleBot(BotAI):
    def __init__(self):
        super().__init__()
        self.stim_used = set()

    async def on_step(self, iteration: int):
        if iteration == 0:
            print("Marine Micro Bot - 5v2 Marines vs Zealots!")
        if self.units.exists:
            await self.marine_micro()

    async def marine_micro(self):
        marines = self.units(UnitTypeId.MARINE)
        zealots = self.enemy_units(UnitTypeId.ZEALOT)

        if not marines.exists or not zealots.exists:
            return

        close_zealots = zealots.filter(lambda z:
            marines.closest_to(z.position).distance_to(z) < 6)

        if close_zealots.exists:
            for marine in marines:
                if (marine.tag not in self.stim_used and
                    AbilityId.EFFECT_STIM_MARINE in
                    await self.get_available_abilities(marine)):
                    marine(AbilityId.EFFECT_STIM_MARINE)
                    self.stim_used.add(marine.tag)

        target = min(zealots, key=lambda z: z.health + z.shield)

        for marine in marines:
            dist = marine.distance_to(target)
            if dist < 1:
                retreat_pos = marine.position.towards(target.position, -3)
                marine.move(retreat_pos)
            elif dist <= 5:
                marine.attack(target)
            else:
                marine.move(target.position)
```

### A.3.2 Intermediate Stage: Multi-unit Coordination

```
class BattleBot(BotAI):
    async def on_step(self, iteration: int):
        if iteration == 0:
            print("Terran Battle Bot Activated!")

        if self.units.exists:
            await self.control_ghosts()
            await self.control_marines()
            await self.control_marauders()
            await self.control_medivacs()

    async def control_ghosts(self):
        ghosts = self.units(UnitTypeId.GHOST)
        if not ghosts.exists:
            return

        high_templars = self.enemy_units(UnitTypeId.HIGHTEMPLAR)
        stalkers = self.enemy_units(UnitTypeId.STALKER)

        for ghost in ghosts:
            if high_templars.exists:
                templar = high_templars.closest_to(ghost.position)
                if ghost.distance_to(templar) < 12:
                    if AbilityId.SNIPE_SNIPE in
                        await self.get_available_abilities(ghost):
                        ghost(AbilityId.SNIPE_SNIPE, templar)
                    elif AbilityId.EMP_EMP in
                        await self.get_available_abilities(ghost):
                        ghost(AbilityId.EMP_EMP, templar.position)
                ghost.move(templar.position)
                if AbilityId.BEHAVIOR_CLOAKON_GHOST in
                    await self.get_available_abilities(ghost):
                    ghost(AbilityId.BEHAVIOR_CLOAKON_GHOST)
            elif stalkers.exists:
                target = stalkers.closest_to(ghost.position)
                ghost.attack(target)

    async def control_medivacs(self):
        medivacs = self.units(UnitTypeId.MEDIVAC)
        if not medivacs.exists:
            return

        bio_units = self.units.filter(lambda unit:
                    unit.type_id in {UnitTypeId.MARINE,
                    UnitTypeId.MARAUDER})
        for medivac in medivacs:
            injured = bio_units.filter(lambda unit:
                    unit.health_percentage < 0.75)
            if injured.exists:
                target = injured.closest_to(medivac.position)
                medivac.move(target.position)
                if medivac.distance_to(target) < 5:
                    medivac(AbilityId.MEDIVACHEAL_HEAL, target)
            elif bio_units.exists:
                medivac.move(bio_units.center)
```

### A.4 Advanced Stage: Complex Multi-unit Tactical Framework

```
class BattleBot(BotAI):
    async def on_step(self, iteration: int):
        if iteration == 0:
```

```
810             self.setup_complete = False
811             await self.initial_positioning()
812             self.setup_complete = True
813
814         bio_units = self.units.of_type({UnitTypeId.MARINE,
            UnitTypeId.MARAUDER})
815         medivacs = self.units(UnitTypeId.MEDIVAC)
816
817         await self.avoid_aoe(bio_units + medivacs)
818         await self.control_siege_tanks()
819         await self.control_vikings()
820         await self.control_liberators()
821         await self.control_ghosts()
822         await self.control_bio(UnitTypeId.MARINE)
823         await self.control_bio(UnitTypeId.MARAUDER)
824         await self.control_medivacs()
825
826     async def avoid_aoe(self, units: Units):
827         storms = [e for e in self.state.effects
828                  if e.id == EffectId.PSISTORMPERSISTENT]
829         disruptor_balls = self.enemy_units(UnitTypeId.DISRUPTORPHASED)
830
831         threats = []
832         for storm in storms:
833             threats.append((storm.position, 2.5))
834         for ball in disruptor_balls:
835             threats.append((ball.position, 2.5))
836
837         for unit in units:
838             for pos, radius in threats:
839                 if unit.distance_to(pos) < radius:
840                     away = unit.position.towards(pos, -3)
841                     unit.move(away)
842                     break
843
844     async def control_siege_tanks(self):
845         siege_tanks = self.units(UnitTypeId.SIEGETANKSIEGED)
846         sieged_tanks = self.units(UnitTypeId.SIEGETANKSIEGED)
847
848         for tank in siege_tanks:
849             if tank.distance_to(Point2((15, 15))) > 2:
850                 continue
851             abilities = await self.safe_get_abilities(tank)
852             if AbilityId.SIEGEMODE_SIEGEMODE in abilities:
853                 tank(AbilityId.SIEGEMODE_SIEGEMODE)
854
855         if sieged_tanks.exists:
856             enemies = self.enemy_units
857             if not enemies.exists:
858                 return
859             priority_targets = enemies.of_type([UnitTypeId.COLOSSUS,
860                                 UnitTypeId.STALKER, UnitTypeId.HIGHTEMPLAR,
861                                 UnitTypeId.ZEALOT])
862
863             for tank in sieged_tanks:
                    targets_in_range = priority_targets.in_attack_range_of(tank)
                    if targets_in_range:
                        target = min(targets_in_range,
                                key=lambda t: (t.type_id not in
                                {UnitTypeId.COLOSSUS, UnitTypeId.HIGHTEMPLAR},
                                t.distance_to(tank)))
                        tank.attack(target)
```

Table 6: Complete curriculum evolution across five independent runs. Each row is one curriculum within a path; acceptance requires $\mathcal{P} \geq 0.67$. Detailed map/unit/tech descriptors for each curriculum are elided for space.

| Path | Task | Agent Composition | Enemy Composition | Result |
|---|---|---|---|---|
| 1 | 1 | Marine (5) | Zealot (2, Charge) | 67% |
| | 2 | Marine (10), Marauder (5), Medivac (1) | Zealot (5, Charge), Stalker (5, Blink), HighTemplar (1, PsiStorm) | 67% |
| | 3 | Marine (15), Marauder (8), Ghost (2), Medivac (2), SiegeTank (1) | Zealot (10, Charge), Stalker (8, Blink), HighTemplar (2, PsiStorm), Colossus (1, ExtLance) | Failed |
| | 4 | Marine (12), Marauder (6), Ghost (1), Medivac (1) | Zealot (8, Charge), Stalker (6, Blink), HighTemplar (1, PsiStorm) | 67% |
| | 5 | Marine (18), Marauder (10), Ghost (2), Medivac (2), SiegeTank (1), Viking (2) | Zealot (12, Charge), Stalker (10, Blink), HighTemplar (2, PsiStorm), Colossus (1, ExtLance) | 67% |
| | 6 | Final Task (Table 1) | Final Task (Table 1) | 100% |
| 2 | 1 | Marine (5) | Zealot (2, Charge) | 100% |
| | 2 | Marine (10), Marauder (5), Medivac (2), SiegeTank (1) | Zealot (8, Charge), Stalker (5, Blink), HighTemplar (2, PsiStorm) | 67% |
| | 3 | Marine (15), Marauder (8), Ghost (2), Medivac (3), SiegeTank (1), Viking (4) | Zealot (12, Charge), Stalker (10, Blink), HighTemplar (3, PsiStorm), Colossus (2, ExtLance) | Failed |
| | 4 | Marine (12), Marauder (6), Ghost (1), Medivac (2), SiegeTank (1), Viking (2) | Zealot (10, Charge), Stalker (8, Blink), HighTemplar (2, PsiStorm), Colossus (1, ExtLance) | Failed |
| 3 | 1 | Marine (5) | Zealot (2, Charge) | 100% |
| | 2 | Marine (8), Marauder (5, PunisherGrenades), SiegeTank (1), Medivac (1, CaduceusReactor) | Zealot (7, Charge), Stalker (3, Blink), HighTemplar (2, PsiStorm), Colossus (1, ExtLance) | Failed |
| | 3 | Marine (8), Marauder (4), SiegeTank (1), Medivac (2, CaduceusReactor) | Zealot (5, Charge), Stalker (2, Blink), Colossus (1) | 100% |
| | 4 | Marine (14), Marauder (7, PunisherGrenades), SiegeTank (2), Medivac (3, CaduceusReactor), Viking (2), Ghost (1) | Zealot (9, Charge), Stalker (5, Blink), HighTemplar (2, PsiStorm), Disruptor (1) | Failed |
| | 5 | Marine (10), Marauder (5), SiegeTank (1), Medivac (2, CaduceusReactor) | Zealot (6, Charge), Stalker (3, Blink), Colossus (1) | 100% |
| | 6 | Marine (14), Marauder (7, PunisherGrenades), SiegeTank (2), Medivac (3, CaduceusReactor), Viking (2), Ghost (1) | Zealot (9, Charge), Stalker (5, Blink), HighTemplar (2, PsiStorm), Colossus (2, ExtLance), Disruptor (1) | Failed |
| | 7 | Marine (14), Marauder (7, PunisherGrenades), SiegeTank (1), Medivac (2, CaduceusReactor), Ghost (1) | Zealot (9, Charge), Stalker (4, Blink), Colossus (1, ExtLance) | Failed |
| 4 | 1 | Marine (5) | Zealot (2, Charge) | 67% |
| | 2 | Marine (10), Marauder (5), Ghost (2), Medivac (1, CaduceusReactor) | Zealot (8, Charge), Stalker (4, Blink), HighTemplar (1, PsiStorm) | 100% |
| | 3 | Marine (15), Marauder (8), Ghost (3), Medivac (2, CaduceusReactor), SiegeTank (1), Viking (2) | Zealot (12, Charge), Stalker (8, Blink), HighTemplar (2, PsiStorm), Colossus (1, ExtLance) | Failed |
| 5 | 1 | Marine (5) | Zealot (2, Charge) | 67% |
| | 2 | Marine (10), Marauder (5), Medivac (1) | Zealot (5, Charge), Stalker (5, Blink), HighTemplar (1, PsiStorm) | 67% |
| | 3 | Marine (15), Marauder (8), Ghost (2), Medivac (2), SiegeTank (1) | Zealot (10, Charge), Stalker (8, Blink), HighTemplar (2, PsiStorm), Colossus (1, ExtLance) | Failed |
| | 4 | Marine (12), Marauder (6), Ghost (1), Medivac (1) | Zealot (8, Charge), Stalker (6, Blink), HighTemplar (1, PsiStorm) | 67% |
| | 5 | Marine (18), Marauder (10), Ghost (2), Medivac (2), SiegeTank (1), Viking (2) | Zealot (12, Charge), Stalker (10, Blink), HighTemplar (2, PsiStorm), Colossus (1, ExtLance) | 67% |
| | 6 | Final Task (Table 1) | Final Task (Table 1) | Failed |

# B COMPLETE CURRICULUM EVOLUTION ACROSS ALL FIVE PATHS (STARCRAFT II)

## B.1 STARCRAFT II MINI GAME MAPS

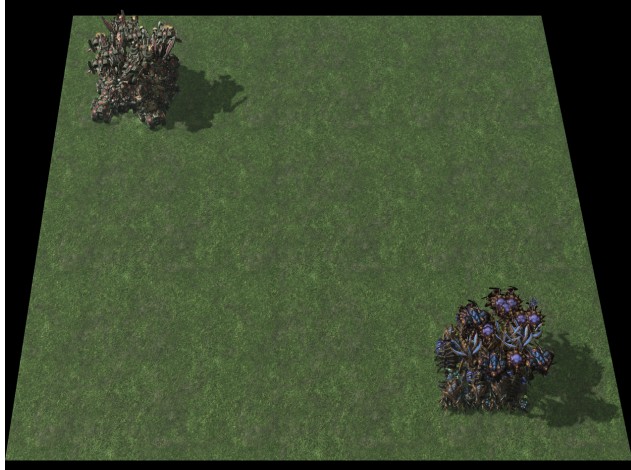

(a) Flat map layout

(b) Terran vs Protoss on Flat

(c) Terran vs Zerg on Flat

Figure 4: Flat map configurations: base layout and unit compositions for different matchups

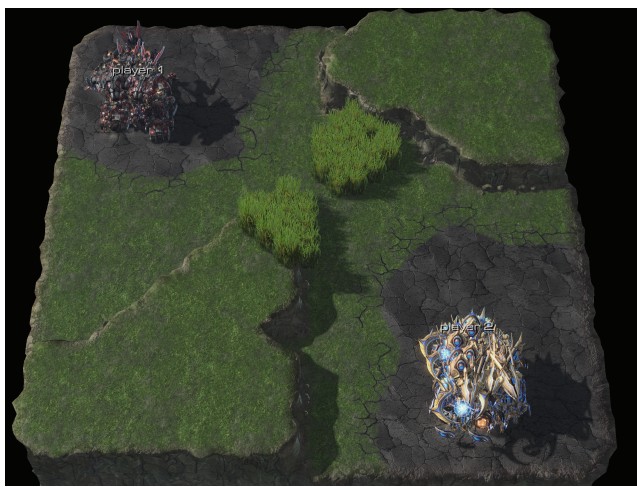

(a) Bush map layout

(b) Terran vs Protoss on Bush

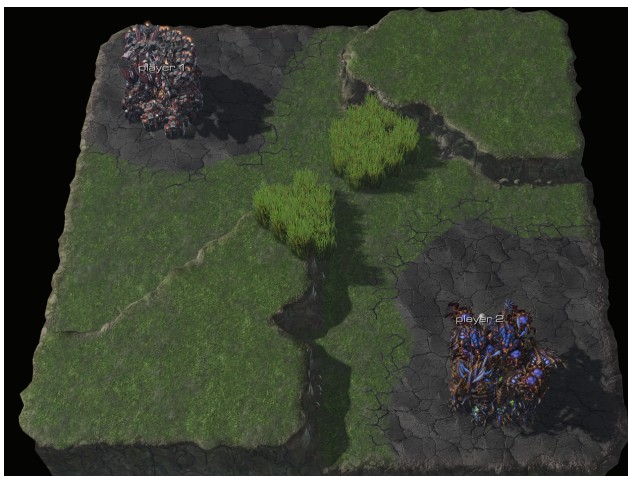

(c) Terran vs Zerg on Bush

Figure 5: Bush map configurations: base layout and unit compositions for different matchups

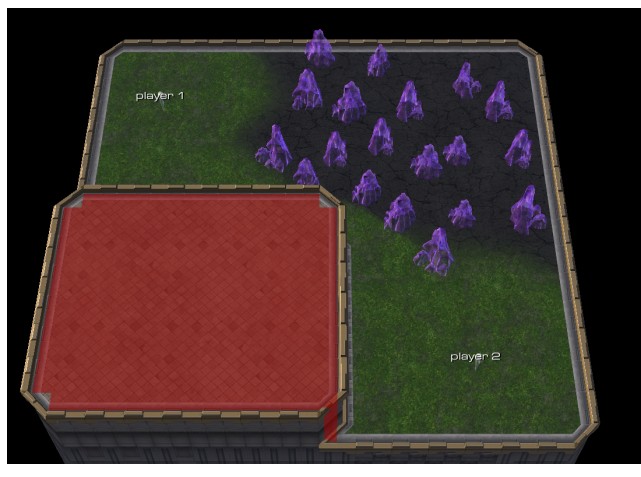

(a) Corner map layout

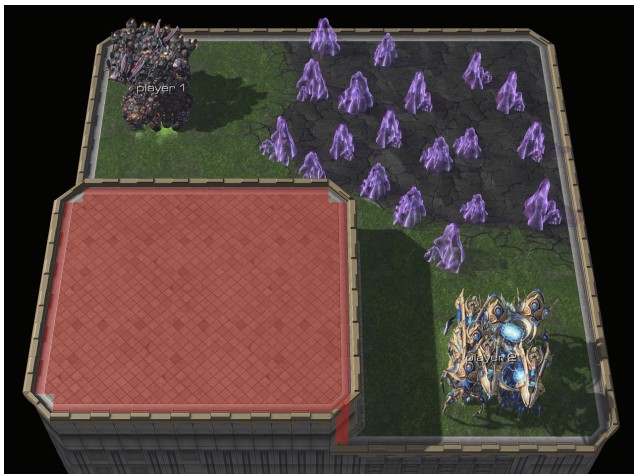

(b) Terran vs Protoss on Corner

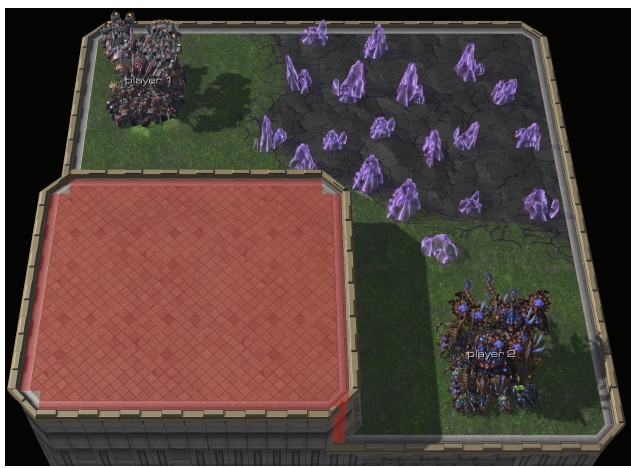

(c) Terran vs Zerg on Corner

Figure 6: Corner map configurations: base layout and unit compositions for different matchups

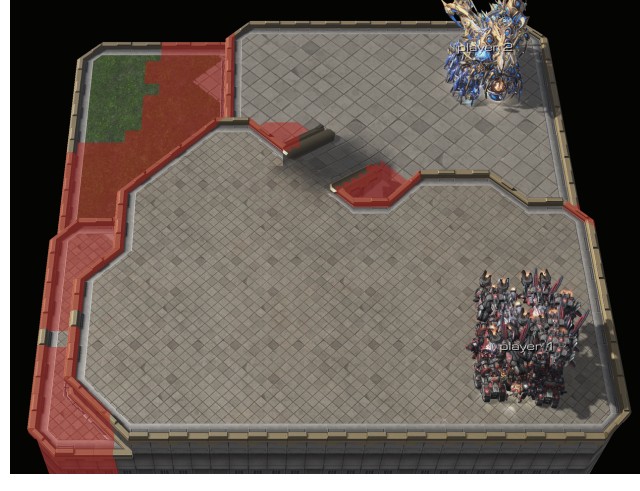

(a) Main map layout

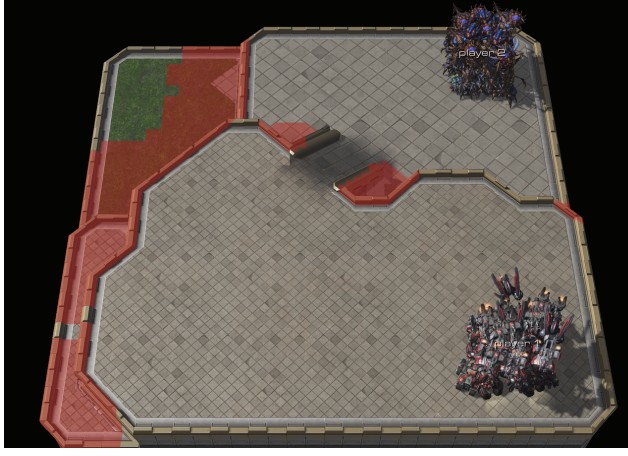

(b) Terran vs Protoss on Main

(c) Terran vs Zerg on Main

Figure 7: Main map configurations: base layout and unit compositions for different matchups

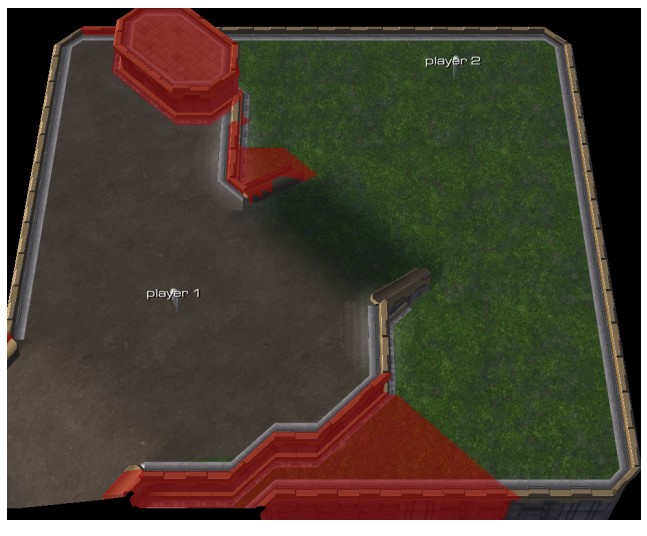

(a) Ramp map layout

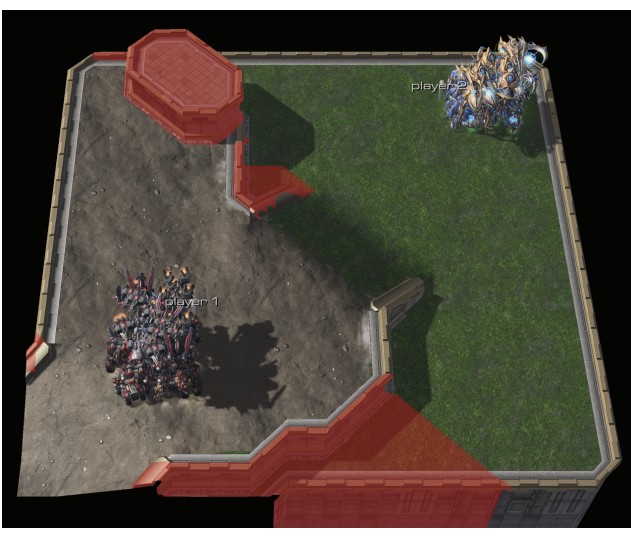

(b) Terran vs Protoss on Ramp

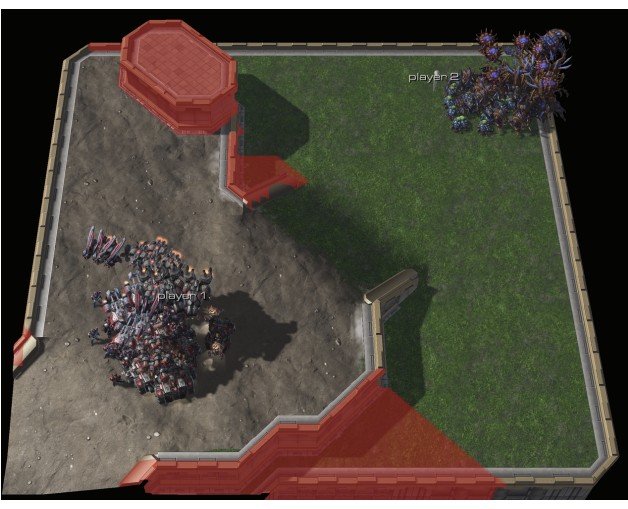

(c) Terran vs Zerg on Ramp

Figure 8: Ramp map configurations: base layout and unit compositions for different matchups

# C  OVERCOOKED DETAILS

## C.1  MARL TRAINING CONFIGURATION

For the Overcooked experiments, we employ the E3T (Efficient End-to-End Training) algorithm (Yan et al., 2023) as our MARL training framework. The training configuration is detailed in Table 7.

Table 7: Overcooked MARL training hyperparameters

| Parameter | Value |
|---|---|
| Algorithm | E3T |
| Number of agents | 2 |
| Episode length | 400 |
| Number of environment steps per curriculum | $10^7$ |
| PPO epochs | 15 |
| Number of mini-batches | 1 |
| Rollout threads | 100 |
| Evaluation threads | 10 |
| Evaluation interval | 20 episodes |
| **Entropy Regularization Schedule** | |
| Entropy coefficients | [0.2, 0.05, 0.01] |
| Entropy coefficient horizons | $[0, 6 \times 10^6, 10^7]$ |
| **Network Architecture** | |
| CNN layers | [(32, 3×3, stride=1), (64, 3×3, stride=1), (32, 3×3, stride=1)] |
| Recurrent policy | LSTM |
| Shared policy | True |
| **E3T Specific** | |
| Epsilon (diversity bonus) | 0.25 |
| Weights copy factor | 0.1 |
| Random index | Enabled |
| **Curriculum-specific** | |
| Reward shaping horizon | $10^8$ |
| Budget per curriculum | $10^7$ timesteps |

## C.2  CURRICULUM DESIGN FOR OVERCOOKED

The curriculum progression in Overcooked is structured around three key dimensions:

1. **Layout Complexity**: Starting from simple layouts (e.g., small_corridor) with direct paths between stations, progressing to complex layouts with obstacles and longer navigation requirements.

2. **Order Complexity**: Beginning with single-ingredient dishes, advancing to multi-ingredient recipes requiring precise coordination between agents.

3. **Temporal Constraints**: Initially allowing unlimited time for order completion, then introducing time pressure and simultaneous order requirements.

The acceptance criterion $\mathcal{P}(\pi|C) = 1$ is achieved when all required orders are completed within the episode. After meeting this criterion, agents can pursue bonus orders to maximize total deliveries. The entropy regularization schedule ensures exploration early in training (high entropy) while converging to more deterministic policies as training progresses.

For layouts with particular navigation challenges (e.g., small_corridor), we adjust the entropy coefficient horizons to $[0, 8 \times 10^6, 10^7]$ to allow for extended exploration before convergence. The shared policy architecture enables agents to learn cooperative behaviors more efficiently by sharing representations while maintaining individual action distributions.

## D   OVERCOOKED EXPERIMENTAL RESULTS DETAILS

This appendix provides detailed experimental results for the Overcooked environment, showing the complete curriculum evolution and performance metrics for two different map configurations.

### D.1   MAP CONFIGURATION 1

#### D.1.1   FINAL LAYOUT SPECIFICATION

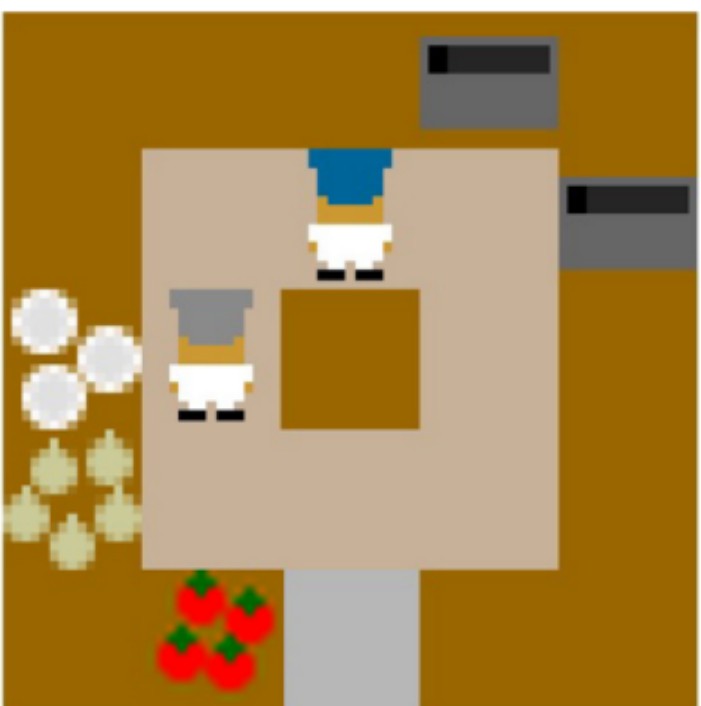

Figure 9: overcook map1

```
{
  "grid": "XXXPX
          X 2 P
          D1X X
          O   X
          XTSXX",
  "start_all_orders": [
    {"ingredients": ["onion", "tomato"]},
    {"ingredients": ["onion", "onion"]},
    {"ingredients": ["onion", "tomato", "tomato"]},
    {"ingredients": ["onion", "onion", "tomato"]},
    {"ingredients": ["onion", "onion", "onion"]}
  ],
  "recipe_value": [10, 10, 20, 20, 20],
  "recipe_time": [10, 10, 20, 20, 20]
}
```

Grid Legend: X=Wall, P=Pot, D=Dish, O=Onion, T=Tomato, S=Service, 1/2=Agent spawn

Table 8: Map 1: Curriculum progression and performance metrics

| Task | Orders | Agent0 Delivery | Agent1 Delivery | Total Delivery | Sparse Reward |
|------|--------|-----------------|-----------------|----------------|---------------|
| Task 0 | 3 | 11.75 | 12.33 | 24.08 | 239.2 |
| Task 1 | 4 | 14.56 | 14.33 | 28.89 | 288.1 |
| Task 2 (Final) | 5 | 14.74 | 14.36 | 29.10 | 290.9 |
| Direct Training | 5 | 11.93 | 12.18 | 24.11 | 240.5 |

### D.1.2 CURRICULUM EVOLUTION RESULTS

### D.1.3 DETAILED PERFORMANCE METRICS COMPARISON

Table 9: Map 1: Key performance indicators for final task

| Metric | EvoCurr (Final) | | Direct Training | |
|--------|-----------------|--------|-----------------|--------|
| | Agent0 | Agent1 | Agent0 | Agent1 |
| Onion Placement in Pot | 15.77 | 15.94 | 13.38 | 13.57 |
| Tomato Placement in Pot | 0.31 | 0.32 | 0.21 | 0.26 |
| Useful Dish Pickup | 7.81 | 7.63 | 6.15 | 6.34 |
| Soup Pickup | 7.68 | 7.55 | 6.31 | 6.33 |
| Cook Actions | 7.44 | 8.67 | 6.34 | 7.30 |
| Delivery Actions | 7.37 | 7.19 | 5.99 | 6.11 |
| Idle Movement | 9.40 | 8.68 | 14.90 | 15.10 |

## D.2 MAP CONFIGURATION 2

### D.2.1 FINAL LAYOUT SPECIFICATION

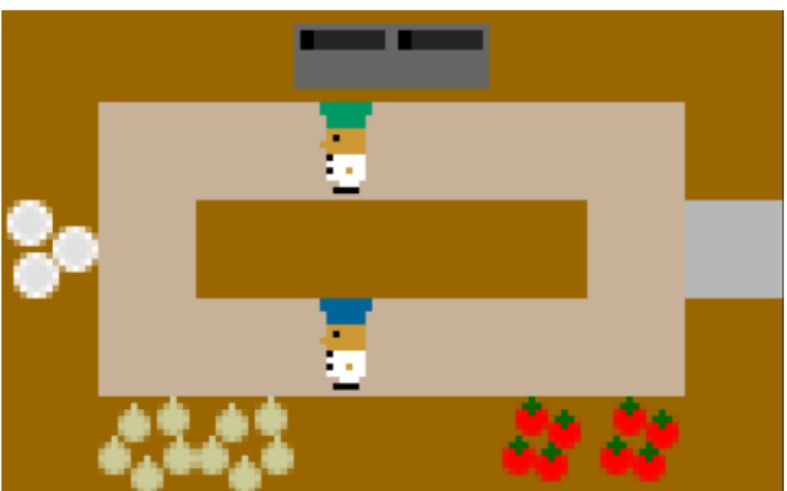

Figure 10: overcook map2

```
{
  "grid": "XXXPDPXXX
          X   2   X
          X XXXXX X
          X   1   X
          XXXOSTXXX",
```

```
  "start_all_orders": [
    {"ingredients": ["onion", "tomato"]},
    {"ingredients": ["onion", "onion"]},
    {"ingredients": ["onion", "tomato", "tomato"]},
    {"ingredients": ["onion", "onion", "tomato"]},
    {"ingredients": ["onion", "onion", "onion"]}
  ],
  "recipe_value": [10, 10, 20, 20, 20],
  "recipe_time": [10, 10, 20, 20, 20]
}
```

Grid Legend: X=Wall, P=Pot, D=Dish, O=Onion, T=Tomato, S=Service, 1/2=Agent spawn

### D.2.2 CURRICULUM EVOLUTION RESULTS

Table 10: Map 2: Curriculum progression and performance metrics

| Task | Orders | Agent0 Delivery | Agent1 Delivery | Total Delivery | Sparse Reward |
|------|--------|-----------------|-----------------|----------------|---------------|
| Task 0 | 3 | 7.28 | 7.25 | 14.53 | 290.3 |
| Task 1 | 5* | 7.32 | 7.28 | 14.60 | 291.6 |
| Task 2 (Final) | 5 | 7.82 | 10.82 | 18.64 | 186.2 |
| Direct Training | 5 | 7.40 | 8.96 | 16.36 | 163.4 |

*Task 1 includes two single-ingredient recipes alongside complex recipes for easier transition

### D.2.3 DETAILED PERFORMANCE METRICS COMPARISON

Table 11: Map 2: Key performance indicators for final task

| Metric | EvoCurr (Final) | | Direct Training | |
|--------|-----------------|--------|-----------------|--------|
| | Agent0 | Agent1 | Agent0 | Agent1 |
| Onion Placement in Pot | 7.91 | 5.95 | 9.85 | 9.29 |
| Tomato Placement in Pot | 3.17 | 3.87 | 0.08 | 0.07 |
| Useful Dish Pickup | 5.45 | 4.20 | 4.32 | 4.81 |
| Soup Pickup | 4.12 | 5.62 | 4.24 | 4.79 |
| Cook Actions | 5.45 | 4.85 | 5.16 | 4.38 |
| Delivery Actions | 3.92 | 5.42 | 3.71 | 4.49 |
| Size-2 Order Delivery | 3.84 | 5.27 | 3.65 | 4.40 |
| Size-3 Order Delivery | 0.07 | 0.14 | 0.05 | 0.08 |
| Idle Movement | 14.42 | 14.21 | 10.49 | 11.28 |

