# OpenReview forum: "EvoCurr: Self-evolving Curriculum with Behavior Code Generation for Complex Decision-making"
_ICLR.cc/2026/Conference — ICLR 2026 Conference Withdrawn Submission_

### Official Review · Reviewer_zrhK · 2025-10-23

**Soundness:** 2
**Presentation:** 1
**Contribution:** 1
**Rating:** 2
**Confidence:** 3

**Summary:**

The paper introduces **EvoCurr**, a self-evolving curriculum learning framework designed to enable Large Language Models (LLMs) to tackle high-complexity decision-making tasks that require long-horizon reasoning and multi-step coordination, where LLMs typically fail. The framework operates as a cooperative two-agent system where an LLM **Designer** generates adaptive task sequences, and a **Solver** produces executable solutions, supporting both open-loop code generation (behavior trees) and closed-loop policy learning (Multi-Agent Reinforcement Learning). A key innovation is the accepted-floor constraint and feasibility gate, which prevent difficulty regression and ensure monotonic skill advancement by only accepting valid tasks above a previously mastered level. EvoCurr was evaluated on StarCraft II micro-management and Overcooked coordination tasks; in StarCraft II, it achieved combat winning rates over 90% in scenarios where direct code generation failed, and in Overcooked, it achieved 20% higher task completion rates compared to direct training, successfully extending LLM capabilities to previously unreachable complex tasks.

**Strengths:**

1. The proposed framework can be applied to a wide range of other tasks.

2. Compared to the baseline proposed in the paper, significant improvements were achieved in both StarCraft and Overcooked tasks.

**Weaknesses:**

1. Overall, the framework appears relatively simple, consisting solely of a designer and a solver. This makes the framework not entirely suitable for all tasks.  The author needs to provide further evidence to demonstrate the applicability of this framework.

2. I have reservations about the novelty of the paper.  This paper designs a curriculumlearning framework and validates its effectiveness in the LLM domain. I personally argue that this framework represents a simplification of current coding agents such as Claude Code, Gemini CLI, and GitHub Copilot Agent. I wonder whether existing coding agents can accomplish this task after integrating with simulation testbeds like SC2. **Please note that Coding Agent is not equivalent to an LLM.**

3. The baseline for comparison needs to be more comprehensive. Currently, it only includes various base models without comparing other curriculum learning algorithms or agentic-type algorithms.

**Questions:**

1. In Section 3.1, how are the difficulty function and distance function defined?

2. Where are the results for GPT-5, Claude 4, and Gemini 2.5 mentioned in Line 323?

typo:
1. Line 324, there may be an extra "." here.
2. Line 355, "scores ¡ 0.5" should be sorces < 0.5？

---

> ### Author Response · Authors · 2025-11-24
> **Answers for questions part1**
>
> We would like to express our sincere gratitude to the reviewer for taking the time to carefully review our manuscript and provide such detailed and constructive feedback. Your thoughtful comments and valuable suggestions have been instrumental in helping us refine our work and address important aspects that strengthen the contribution. We have thoroughly considered each of your points and provided detailed responses below.
>
> First, to preserve EvoCurr's generalization capability, both the difficulty function d(C) and distance function are autonomously determined by the LLM itself rather than being manually pre-defined. We simply specify the requirements for these functions in the prompt (e.g., ensuring monotonic difficulty increase), but the actual assessment and computation are performed by the LLM's reasoning capabilities. The LLM evaluates difficulty values and distances between curricula based on its understanding of task complexity and curriculum progression. This design choice is fundamental to EvoCurr's generalization by delegating these assessments to the LLM rather than hardcoding scenario-specific metrics, so the framework can adapt across different domains (SC2, Overcooked, etc.) without requiring manual reconfiguration.
>
> Second, we sincerely appreciate your careful review and attention to detail in identifying the typos and errors in our manuscript. Your meticulous feedback helps us improve the overall quality and readability of our work. And the results will be presented below (we have added performance results for coding agents, such as Claude Code, Codex, and Cursor, but Cursor's performance was too poor to generate correct results):
>
> ## Score Table
>
> | Task | EvoCurr (DeepSeek v3.1) | EvoCurr (Claude-4) | EvoCurr (GPT-5) | DeepSeek | GPT-5 | Claude-4 | Gemini-2.5 | Claude_code | Codex | Cursor |
> | :--- | :---: | :---: | :---: | :---: | :---: | :---: | :---: | :---: | :---: | :---: |
> | bush (TVP) | **0.699841** | 0.636 | 0.696 | 0.338 | 0.436 | 0.357 | 0.499 | 0.194 | 0.279 | 0 |
> | corridor (TVP) | **0.714642** | 0.610 | 0.620 | 0.561 | 0.516 | 0.638 | 0.264 | 0.192 | 0.213 | 0 |
> | corner (TVP) | 0.551 | 0.637 | 0.659 | **0.666** | 0.526 | 0.625 | 0.559 | 0.282 | 0.275 | 0 |
> | flat (TVP) | **0.699631** | 0.607 | 0.629 | 0.412 | 0.567 | 0.390 | 0.308 | 0.216 | 0.221 | 0 |
> | main (TVP) | 0.728 | 0.657 | 0.667 | 0.697 | 0.632 | 0.677 | **0.774** | 0.343 | 0.408 | 0 |
> | ramp (TVP) | 0.723 | 0.635 | 0.681 | 0.457 | 0.484 | **0.745** | 0.662 | 0.218 | 0.239 | 0 |
> | bush (TVZ) | **0.732238** | 0.670 | 0.647 | 0.612 | 0.342 | 0.647 | 0.368 | 0.465 | 0.318 | 0 |
> | corridor (TVZ) | **0.697976** | 0.650 | 0.575 | 0.588 | 0.569 | 0.656 | 0.598 | 0.343 | 0.313 | 0 |
> | corner (TVZ) | **0.840462** | 0.742 | 0.621 | 0.415 | 0.698 | 0.776 | 0.801 | 0.741 | 0.621 | 0 |
> | flat (TVZ) | **0.626284** | 0.612 | 0.531 | 0.361 | 0.297 | 0.493 | 0.377 | 0.311 | 0.278 | 0 |
> | main (TVZ) | 0.702 | **0.760** | 0.753 | 0.685 | 0.433 | 0.431 | 0.668 | 0.455 | 0.451 | 0 |
> | ramp (TVZ) | **0.746564** | 0.652 | 0.706 | 0.450 | 0.454 | 0.682 | 0.294 | 0.454 | 0.293 | 0 |
>
> ---
>
> ## Winning Rate Table (%)
>
> | Task | EvoCurr (DeepSeek v3.1) | EvoCurr (Claude-4) | EvoCurr (GPT-5) | DeepSeek | GPT-5 | Claude-4 | Gemini-2.5 | Claude_code | Codex |  Cursor |
> | :--- | :---: | :---: | :---: | :---: | :---: | :---: | :---: | :---: | :---: | :---: |
> | bush (TVP) | **100** | **100** | **100** | 0 | 0 (equal 3 times) | 10 | 0 (equal 4 times) | 0 | 0 | 0 |
> | corridor (TVP) | **100** | 70 (equal 3 times) | 90 (equal 1 times) | 80 | 80 (equal 2 times) | **100** | 0 (equal 2 times) | 0 | 0 | 0 |
> | corner (TVP) | 90 | **100** | **100** | **100** | 90 | **100** | 30 | 0 | 0 | 0 |
> | flat (TVP) | **100** | **100** | **100** | 0 | **100** | 40 | 0 | 0 | 0 | 0 |
> | main (TVP) | **100** | **100** | **100** | **100** | **100** | **100** | **100** | 0 (equal 2 times) | 0 | 0 |
> | ramp (TVP) | **100** | **100** | **100** | 10 | 0 (equal 4 times) | **100** | **100** | 0 | 0 | 0 |
> | bush (TVZ) | **100** | 90 (equal 1 times) | **100** | **100** | 0 | **100** | 0 | 40 (equal 1 times) | 0 (equal 5 times) | 0 |
> | corridor (TVZ) | **100** | **100** | 90 | 90 | 0 (equal 4 times) | **100** | **100** | 0 (equal 2 times) | 0 (equal 4 times) | 0 |
> | corner (TVZ) | **100** | **100** | **100** | 30 | **100** | **100** | **100** | **100** | **100** | 0 |
> | flat (TVZ) | **100** | **100** | 90 | 0 | 0 | 0 | 0 | 0 | 0 | 0 |
> | main (TVZ) | **100** | **100** | **100** | **100** | 0 | 0 | **100** | 10 (equal 2 times) | 20 (equal 2 times) | 0 |
> | ramp (TVZ) | **100** | **100** | **100** | 50 | 60 (equal 1 times) | **100** | 0 | 20 (equal 5 times) | 0 | 0 |
>
> ---
>
> **Additionally:** We tested Cursor with Grok4, but it consistently generated incorrect attribute or function names.

---

> ### Author Response · Authors · 2025-11-24
> **Answers for questions part2**
>
> Third, the advantages of EvoCurr over agentic-type algorithms are already evident from the table above. Meanwhile, regarding curriculum learning algorithms, most curriculum learning algorithms lack generalizability and instead rely on manual curricula design for specific scenarios. However, EvoCurr can automatically construct curricula based solely on the given final task, eliminating the need for the tedious and complex curriculum design required by other curriculum learning methods. It can also verify curriculum effectiveness through the Solver and automatically adjust task difficulty. Furthermore, it can be extended to more experimental scenarios by simply modifying the experimental scenario settings, which ensures EvoCurr's generalizability.
>
> Finally, we sincerely appreciate the time and effort you have dedicated to reviewing our work. Your insightful feedback and constructive suggestions have been invaluable in enhancing the quality of our manuscript. We believe the revisions made in response to your comments have significantly strengthened our contribution. If there are any remaining concerns or aspects that require further clarification, we welcome the opportunity for continued discussion. Thank you again for your careful consideration and valuable guidance.

---

> > ### Comment · Reviewer_zrhK · 2025-11-26
> >
> > Thank you for the detailed rebuttal and the additional experimental results. I appreciate the authors’ efforts in clarifying aspects of the framework and providing more performance tables. However, after carefully evaluating the response, several core concerns remain unaddressed, which I summarize below.
> >
> > ### 1. On the construction and generality of the difficulty function (d(C))
> >
> > The rebuttal emphasizes that (d(C)) and the curriculum distance are “determined by the LLM itself” rather than manually defined. However, this does not fully address my original concern. My question is not whether the values are manually specified, but whether the *mechanism* for constructing (d(C)) is sufficiently general to extend beyond the specific domains evaluated.
> >
> > ### 2. Comparison to modern Coding Agents and missing evaluation details
> >
> > My second concern relates to novelty. I view the primary novelty claim of the paper as enabling an LLM to self-design curricula to progressively solve a hard task. My skepticism comes from the observation that modern Coding Agents (e.g., Claude Code, Gemini Code Assist, GitHub Copilot Agent) are already capable of iterative planning, simulation-based refinement, and multi-step debugging under appropriate prompting.
> >
> > While the rebuttal provides single-shot performance results for Claude Code, Codex, and Cursor, the comparison still lacks critical evaluation details:
> >
> > - What prompt templates were used for Coding Agents?
> > - Were these agents asked to perform iterative curriculum-style refinement (from easy to hard)?
> > - Were agent modes or tool-use modes enabled?
> > - Were simulation APIs exposed so they could run code, observe feedback, and adjust solutions?
> > - Were they instructed to construct tasks of increasing difficulty?
> >
> > Without these details, the comparison is inconclusive. My concern is that EvoCurr may correspond to a *specialized prompting pattern* for a Coding Agent rather than a fundamentally new framework. The rebuttal does not convincingly rule out the possibility that existing Coding Agents, when given access to the same simulation interface and a curriculum-style prompting scheme, could achieve similar results.
> >
> > ### 3. Generalization to tasks without explicit difficulty structure
> >
> > The rebuttal argues that EvoCurr generalizes across SC2 and Overcooked. However, both domains have relatively clear, explicit difficulty parameters (unit counts, tech levels, map layouts, number of orders, etc.). Even in these structured domains, the difficulty progression fundamentally relies on the Designer’s ability to reason about these domain-specific knobs.
> >
> > My concern is whether the framework can generalize to tasks where difficulty is not naturally defined or decomposable—for example, daily-life manipulation tasks in Behavior-1K [1], multi-step household planning, or complex embodied tasks without clear parametric difficulty dimensions. It is unclear whether the Designer could construct stable curricula in such cases without significant human-designed scaffolding in the prompts. This again relates to the broader question of whether the mechanism is genuinely general or heavily dependent on domain-specific prompt design.
> >
> >
> > Therefore, my original assessment remains largely unchanged.
> >
> > [1] BEHAVIOR-1K: A Human-Centered, Embodied AI Benchmark with 1,000 Everyday Activities and Realistic Simulation

---

> > > ### Author Response · Authors · 2025-12-01
> > > **Appreciation for Your Feedback and Expression of Our Perspective part2**
> > >
> > > We do the additional experiment by using Claude Code to design tasks and modify code.
> > >
> > > **The result is: (Curriculums are designed by Claude Code)**
> > >
> > > | Curriculum | Attempts | result | Winning Rate |
> > > |-------|----------|------|------|
> > > | Curriculum1 | 1 | 5W/0D/0L | **100%**  |
> > > | Curriculum 2 | 1 | 5W/0D/0L | **100%**  |
> > > | Curriculum 3 | 1 | 5W/0D/0L | **100%**  |
> > > | Curriculum 4 | 1 | 5W/0D/0L | **100%**  |
> > > | Curriculum 5 | 1 | 5W/0D/0L | **100%**  |
> > > | **Final Task** | 2 | 0W/1D/9L | **0%**  |
> > >
> > > **Final winning rate: 0%**
> > >
> > > **Total time**: ~90mins
> > >
> > > **Testing times**: ~100 matches
> > >
> > > After 90 mins runing and over 100 matches, we find that claude code still can not complete the Final task. Even though the generating code had **passed the all Curriculum designed by Claude Code. It couln't win the Final Task even once.**

---

> > > ### Author Response · Authors · 2025-12-01
> > > **Appreciation for Your Feedback and Expression of Our Perspective part5**
> > >
> > > **More details can be checked below:**
> > >
> > > **4. Final Task content:**
> > >
> > > **OUR_UNITS:**
> > >
> > >   Marine, 20, (5,25), Stimpack
> > >
> > >   Marauder, 12, (5,25), Stimpack
> > >
> > >   Medivac, 4, (5,25), Heal
> > >
> > >   Ghost, 8, (5,25), PersonalCloaking
> > >
> > >   SiegeTank, 6, (5,25), SiegeTech
> > >
> > >   VikingFighter, 8, (5,25), AssaultMode
> > >
> > >   Cyclone, 8, (5,25), LockOn
> > >
> > >   WidowMine, 8, (5,25), Burrow
> > >
> > >   Raven, 3, (5,25), HunterSeeker
> > >
> > >   Liberator, 4, (5,25), DefenderMode
> > >
> > >  **ENEMIES_UNITS :**
> > >
> > >   Zealot, 15, (25,5), Charge
> > >
> > >   Stalker, 15, (25,5), BlinkTech
> > >
> > >   Sentry, 10, (25,5), ForceField
> > >
> > >   HighTemplar, 9, (25,5), PsiStormTech
> > >
> > >   Colossus, 4, (25,5), ExtendedThermalLance
> > >
> > >   Tempest, 5, (25,5), GroundAttack
> > >
> > >   Disruptor, 4, (25,5), PurificationNova
> > >
> > >   Carrier, 4, (25,5), InterceptorLaunch
> > >
> > > ---
> > >
> > > **5. Experiment Process:**
> > >
> > > Timeline Overview (A simple sample expresses the process of testing):
> > >
> > > | Phase| Content |
> > > |------|------|
> > > | Generate Code and Debug Phase | Generate the control Code and fixed dependencies, win/loss detection bugs |
> > > | First Test Round | Testing the code. |
> > > | Code Optimization | Rewrote battle code and improve. |
> > > | Second Test Round | New code generate and test when it hard to pass the designed Curriculum. Generating the code from the start. |

---

> > > ### Author Response · Authors · 2025-12-01
> > > **Appreciation for Your Feedback and Expression of Our Perspective part6**
> > >
> > > **6. Battle Code Sample:**
> > > ```
> > > class BattleBot(BotAI):
> > >     """
> > >     Terran Auto-Micro Bot - Optimized Version
> > >     Strategy: Immediate Stimpack + Focus Fire Nearest Enemy + Simple Kiting + Medivac Healing
> > >     """
> > >
> > >     def __init__(self):
> > >         super().__init__()
> > >         self.stim_used = set()
> > >         self.started = False
> > >
> > >     async def on_step(self, iteration: int):
> > >         # Get units
> > >         marines = self.units(UnitTypeId.MARINE)
> > >         marauders = self.units(UnitTypeId.MARAUDER)
> > >         medivacs = self.units(UnitTypeId.MEDIVAC)
> > >         ghosts = self.units(UnitTypeId.GHOST)
> > >         tanks = self.units(UnitTypeId.SIEGETANK) | self.units(UnitTypeId.SIEGETANKSIEGED)
> > >
> > >         bio_units = marines | marauders
> > >         all_combat = bio_units | ghosts | tanks
> > >         enemies = self.enemy_units
> > >         enemy_pos = Point2((27, 27))
> > >
> > >         if not all_combat.exists:
> > >             return
> > >
> > >         # No enemies - quick attack
> > >         if not enemies.exists:
> > >             for unit in all_combat:
> > >                 unit.attack(enemy_pos)
> > >             for medivac in medivacs:
> > >                 medivac.move(enemy_pos)
> > >             return
> > >
> > >         # === Optimization 1: Stimpack immediately upon first enemy sight ===
> > >         if not self.started:
> > >             self.started = True
> > >             for unit in bio_units:
> > >                 if unit.health > 20:
> > >                     if unit.type_id == UnitTypeId.MARINE:
> > >                         unit(AbilityId.EFFECT_STIM_MARINE)
> > >                     elif unit.type_id == UnitTypeId.MARAUDER:
> > >                         unit(AbilityId.EFFECT_STIM_MARAUDER)
> > >                     self.stim_used.add(unit.tag)
> > >
> > >         # === Optimization 2: Simplified target selection ===
> > >         low_health = enemies.filter(lambda u: u.health_percentage < 0.4)
> > >         if low_health.exists:
> > >             target = low_health.closest_to(bio_units.center)
> > >         else:
> > >             target = enemies.closest_to(bio_units.center)
> > >
> > >         # === Optimization 3: Kiting tactics ===
> > >         for unit in bio_units:
> > >             closest = enemies.closest_to(unit.position)
> > >             dist = unit.distance_to(closest)
> > >             # Maintain distance from melee units
> > >             if closest.type_id in [UnitTypeId.ZEALOT, UnitTypeId.ZERGLING]:
> > >                 if dist < 3:
> > >                     retreat_pos = unit.position.towards(closest.position, -4)
> > >                     unit.move(retreat_pos)
> > >                     continue
> > >             unit.attack(target)
> > >
> > >         # === Ghost EMP ===
> > >         for ghost in ghosts:
> > >             shielded = enemies.filter(lambda u: u.shield > 20)
> > >             if shielded.exists and ghost.energy >= 75:
> > >                 emp_target = max(shielded, key=lambda u: enemies.closer_than(3, u).amount)
> > >                 ghost(AbilityId.EMP_EMP, emp_target.position)
> > >             else:
> > >                 ghost.attack(target)
> > >
> > >         # === Tank Control ===
> > >         for tank in tanks:
> > >             if tank.type_id == UnitTypeId.SIEGETANK:
> > >                 closest = enemies.closest_to(tank.position)
> > >                 dist = tank.distance_to(closest)
> > >                 if 3 < dist < 13:
> > >                     tank(AbilityId.SIEGEMODE_SIEGEMODE)
> > >                 else:
> > >                     tank.attack(target)
> > >             else:  # SIEGETANKSIEGED
> > >                 if tank.distance_to(enemies.closest_to(tank)) < 3:
> > >                     tank(AbilityId.UNSIEGE_UNSIEGE)
> > >                 else:
> > >                     tank.attack(target)
> > >
> > >         # === Medivac Healing ===
> > >         for medivac in medivacs:
> > >             if bio_units.exists:
> > >                 injured = bio_units.filter(lambda u: u.health_percentage < 0.8)
> > >                 if injured.exists:
> > >                     heal_target = min(injured, key=lambda u: u.health)
> > >                     medivac(AbilityId.MEDIVACHEAL_HEAL, heal_target)
> > >                 else:
> > >                     medivac.move(bio_units.center)
> > > ```

---

> > > ### Author Response · Authors · 2025-12-01
> > > **Appreciation for Your Feedback and Expression of Our Perspective part7**
> > >
> > > **7. Experiment Results**
> > >
> > > First Test Round:
> > > | Curriculum | Attempts | Result | winning rate |
> > > |-------|----------|------|------|
> > > | Curriculum 1 | 1 | 5W/0D/0L | **100%**  |
> > > | Curriculum 2 | 3 | 4W/11D/0L | **27%**  |
> > >
> > > Curriculum 2 Failure Details:
> > > - Attempt 1: 2W/3D/0L (40%)
> > > - Attempt 2: 1W/4D/0L (20%)
> > > - Attempt 3: 1W/4D/0L (20%)
> > >
> > > Second Test Round:
> > > | Curriculum | Attempts | result | Winning Rate |
> > > |-------|----------|------|------|
> > > | Curriculum1 | 1 | 5W/0D/0L | **100%**  |
> > > | Curriculum 2 | 1 | 5W/0D/0L | **100%**  |
> > > | Curriculum 3 | 1 | 5W/0D/0L | **100%**  |
> > > | Curriculum 4 | 1 | 5W/0D/0L | **100%**  |
> > > | Curriculum 5 | 1 | 5W/0D/0L | **100%**  |
> > > | **Final Task** | 2 | 0W/1D/9L | **0%**  |
> > >
> > > Even though the generating code have **pass the all Curriculum** designed by Code Agent, It still **can't to win the Final Task**
> > > ***
> > >
> > > **8. Conclusion**
> > >
> > > Our experiments result shows that **Code agent** like Claude code opus 4.5 **can not handle such difficult promblems by itself.**
> > >
> > > You can clearly observe that Claude Code's performance is extremely poor, achieving only a **0% winning rate** on Ramp, **one of the simplest maps in our task**. This demonstrates that our architecture is effective and reasonable, with capabilities far superior to those of Coding Agents.
> > > ***
> > >
> > > **Thank you again for your questions and detailed feedback, and we hope our responses can address your concerns and help you form a more accurate assessment of our work.**

---

> ### Author Response · Authors · 2025-12-01
> **Appreciation for Your Feedback and Expression of Our Perspective part1**
>
> We thank you for your constructive feedback and follow-up questions. We are happy to address these points below.
>
> >**Q1**: On the construction and generality of the difficulty function (d(C))
>
> The rebuttal emphasizes that (d(C)) and the curriculum distance are “determined by the LLM itself” rather than manually defined. However, this does not fully address my original concern. My question is not whether the values are manually specified, but whether the mechanism for constructing (d(C)) is sufficiently general to extend beyond the specific domains evaluated.
>
> **R1**: Regarding the construction of d(C) in our work, we employ a straightforward approach by incorporating a simple instruction in the prompt:
>
> **"Now you should design curriculum based on the current curriculum, let the curriculum become more close to the final task. But you must make sure this task is banlance."**
>
> This minimalist design intentionally avoids introducing numerous domain-specific parameters or constraints, thereby preserving the framework's generalization capability across different domains.
>
> >**Q2**: Comparison to modern Coding Agents and missing evaluation details
>
> While the rebuttal provides single-shot performance results for Claude Code, Codex, and Cursor, the comparison still lacks critical evaluation details
>
> **R2**: We present the comparison to modern Coding Agents more detailed, and we will show them at the end of answer which contain the content that we design the framework which designs tasks from easy to difficult and resolves them only using Claude Code.
>
> From our experiment, you can clearly observe that Claude Code's performance is extremely poor. And this demonstrates that our architecture is effective and reasonable, with capabilities far superior to those of Coding Agents.
>
> >**Q3**: Generalization to tasks without explicit difficulty structure
>
> whether the framework can generalize to tasks where difficulty is not naturally defined or decomposable—for example, daily-life manipulation tasks in Behavior-1K [1]
>
> **R3**: Thank you very much for raising the point about solving problems that are not naturally defined or decomposable. Since our focus is on addressing curriculum learning scenarios where difficulty can be explicitly defined, we did not extensively discuss solutions for this type of problem. For such problems, we can extend the model to allow the Designer to generate multiple curricula at once, and through the gradients computed during the subsequent Solver process, define the difficulty of the problems and the degree of improvement in effectiveness, and select the curriculum that yields the greatest improvement in results. This approach can also effectively solve problems that are not naturally defined or decomposable. Like the method using in Automatic Curriculum Learning with Gradient Reward Signals (https://arxiv.org/abs/2312.13565)

---

> ### Author Response · Authors · 2025-12-01
> **Appreciation for Your Feedback and Expression of Our Perspective part3**
>
> **More details can be checked following:**
>
> >**Q: What prompt templates were used for Coding Agents**
>
> **1. Prompt template:**
>
> You have a sc2 control task as following:
>
> AGENTS:
>
> Marine, 20, (5,25), Stimpack
>
> Marauder, 12, (5,25), Stimpack
>
> Medivac, 4, (5,25), Heal
>
> Ghost, 8, (5,25), PersonalCloaking
>
> SiegeTank, 6, (5,25), SiegeTech
>
> VikingFighter, 8, (5,25), AssaultMode
>
> Cyclone, 8, (5,25), LockOn
>
> WidowMine, 8, (5,25), Burrow
>
> Raven, 3, (5,25), HunterSeeker
>
> Liberator, 4, (5,25), DefenderMode
>
> ENEMIES:
>
> Zealot, 15, (25,5), Charge
>
> Stalker, 15, (25,5), BlinkTech
>
> Sentry, 10, (25,5), ForceField
>
> HighTemplar, 9, (25,5), PsiStormTech
>
> Colossus, 4, (25,5), ExtendedThermalLance
>
> Tempest, 5, (25,5), GroundAttack
>
> Disruptor, 4, (25,5), PurificationNova
>
> Carrier, 4, (25,5), InterceptorLaunch
>
> You should design the curriculums from easy to hard to make sure the final control code can win the final task.
>
> You can change enviroments by assigning the scenario_type and map_name in rollout_config.py.
>
> Generating the control code and testing them in res-temp.py. You'd better just change the BattleBot class. And you can get
> feedback from the result.
>
> When you pass the current curriculum with a high winning rate, you can design the next curriculum.

---

> ### Author Response · Authors · 2025-12-01
> **Appreciation for Your Feedback and Expression of Our Perspective part4**
>
> > **Q1 & Q2 : Were these agents asked to perform iterative curriculum-style refinement (from easy to hard)** and **Were they instructed to construct tasks of increasing difficulty**
>
> **2. Curriculum Generation Details:**
>
> All curriculums are generated by Claude Code from simple to difficult. And the following is one sample showing the process of curriculum generation.
>
> **Curriculum 1**
>
> Our_units:
>
> | Unit | Count | Position | Upgrades |
> |------|------|------|------|
> | Marine | 20 | (7,7) | Stimpack |
> | Marauder | 12 | (7,7) | Stimpack |
> | Medivac | 4 | (7,7) | Heal |
>
> Enemies_units:
>
> | Unit | Count | Position | Upgrades |
> |------|------|------|------|
> | Zealot | 6 | (27,27) | Charge |
> | Stalker | 4 | (27,27) | BlinkTech |
>
> ---
>
> **Curriculum 2**
>
> Our_units:
>
> | Unit | Count | Position | Upgrades |
> |------|------|------|------|
> | Marine | 20 | (7,7) | Stimpack |
> | Marauder | 12 | (7,7) | Stimpack |
> | Medivac | 4 | (7,7) | Heal |
>
> Enemies_units:
>
> | Unit | Count | Position | Upgrades |
> |------|------|------|------|
> | Zealot | 10 | (27,27) | Charge |
> | Stalker | 8 | (27,27) | BlinkTech |
> ---
>
> **Curriculum 3**
>
> Our_units:
>
> | Unit | Count | Position | Upgrades |
> |------|------|------|------|
> | Marine | 20 | (7,7) | Stimpack |
> | Marauder | 12 | (7,7) | Stimpack |
> | Medivac | 4 | (7,7) | Heal |
>
> Enemies_units:
>
> | Unit | Count | Position | Upgrades |
> |------|------|------|------|
> | Zealot | 12 | (27,27) | Charge |
> | Stalker | 10 | (27,27) | BlinkTech |
> | Sentry | 4 | (27,27) | ForceField |
>
> ---
>
> **Curriculum 4**
>
> Our_units:
>
> | Unit | Count | Position | Upgrades |
> |------|------|------|------|
> | Marine | 20 | (7,7) | Stimpack |
> | Marauder | 12 | (7,7) | Stimpack |
> | Medivac | 4 | (7,7) | Heal |
> | Ghost | 4 | (7,7) | PersonalCloaking |
>
> Enemies_units:
>
> | Unit | Count | Position | Upgrades |
> |------|------|------|------|
> | Zealot | 12 | (27,27) | Charge |
> | Stalker | 10 | (27,27) | BlinkTech |
> | HighTemplar | 4 | (27,27) | PsiStormTech |
> ---
>
> **Curriculum 5**
>
> Our_units:
>
> | Unit | Count | Position | Upgrades |
> |------|------|------|------|
> | Marine | 20 | (7,7) | Stimpack |
> | Marauder | 12 | (7,7) | Stimpack |
> | Medivac | 4 | (7,7) | Heal |
> | Ghost | 6 | (7,7) | PersonalCloaking |
> | SiegeTank | 4 | (7,7) | SiegeTech |
>
> Enemies_units:
>
> | Unit | Count | Position | Upgrades |
> |------|------|------|------|
> | Zealot | 15 | (27,27) | Charge |
> | Stalker | 12 | (27,27) | BlinkTech |
> | HighTemplar | 6 | (27,27) | PsiStormTech |
> | Colossus | 2 | (27,27) | ExtendedThermalLance |
>
> ***
> > **Q3: Were agent modes or tool-use modes enabled**
>
> Yes, they can use anything they want.
>
> > **Q4: Were simulation APIs exposed so they could run code, observe feedback, and adjust solutions**
>
> Yes, the following are the details. It uses the previous result to improve code quality.
>
> 3. Debug Phase Detailed Record:
>
> There are many error accured during the processing. And we are giving one example to show how it like.
>
> Example:
>
> Problem: Original code detected "Result for player 2" instead of "Result for player 1"
>
> Incorrect Code:
>
> Wrong: Detecting opponent (Player 2) result
> ```
> if "Result for player 2" in line:
>     if "Victory" in line:
>         wins += 1
> ```
> Fixed Code:
>
> Correct: Detecting our (Player 1) result
> ```
> if "Result for player 1" in line:
>     if "Victory" in line:
>         wins += 1
>     elif "Defeat" in line:
>         losses += 1
> ```

---

### Official Review · Reviewer_2tCQ · 2025-10-31

**Soundness:** 2
**Presentation:** 2
**Contribution:** 1
**Rating:** 4
**Confidence:** 4

**Summary:**

This paper introduces a curriculum learning method that helps LLMs solve complex, long-horizon tasks by progressively increasing task difficulty in a semi-automated manner. The proposed method, EvoCurr, consists of a designer that adaptively proposes new tasks while a solver produces solutions via behavior-tree code generation or multi-agent RL. Curriculum generation enforces two rules: the accepted-floor constraint, to avoid regressing below mastered difficulty levels, and a feasibility gate, to filter invalid tasks based on success rate. Experiments on StarCraft II micro-management and Overcooked coordination demonstrate that EvoCurr can significantly increase success rates compared to the baselines.

**Strengths:**

- The paper is easy to follow and well-written. The figures explain EvoCurr's components clearly, and the experimental setup and results are well demonstrated too.
- The designer-solver setup is a modular concept that connects curriculum learning to LLM-based solution generation.
- Using two difficult domains as examples for open-loop and closed-loop settings shows that the framework is general in certain aspects.
- Empirical evidence shows that EvoCurr has benefits over direct baselines.

**Weaknesses:**

- Although described as 'self-evolving', the method is not so automated as the difficult levels are pre-defined and do not depend on agent capabilities, but rather heuristically picked properties. So I'd call EvoCurr semi-automated.
- The proposed framework does not seem as novel and generalizable as it is described, as it is a heuristic adoption of existing ideas into two particular domains. For example, the enforced rules sound similar to return and task similarity constraints in self-paced learning, but they are less general.
- No ablation study is carried out.

**Questions:**

- What do you mean by self-evolving, if what and how it can evolve is already constrained based on pre-determined difficulty levels?
- Which rules affect the performance the most? An ablation study looking into this may be nice.
- How would EvoCurr generalize to domains where task similarity cannot be measured based on such properties?

---

> ### Author Response · Authors · 2025-11-24
> **Answers for questions**
>
> We sincerely thank for your valuable time, thoughtful feedback, and constructive suggestions. Your insightful comments have helped us identify areas for improvement and have significantly strengthened our work. We have carefully addressed each concern raised and made corresponding revisions to the manuscript. Below, we provide detailed responses to your comments point by point.
>
> First, We appreciate your concern about level of automation for EvoCurr. We'd like to clarify that d(C) is autonomously controlled by the LLM itself, not manually pre-defined. The LLM evaluates and assigns difficulty values based on its understanding of task complexity、. This design actually reduces reliance on manual heuristics by delegating difficulty assessment to the LLM's reasoning capabilities, thereby preserving generalization. Meanwhile, regarding the acceptance threshold and final task specification, without clear specification, the LLM's potential cannot be fully leveraged, and curriculum design becomes susceptible to unreasonable scenarios. For example, in SC2, the Designer might generate extremely imbalanced scenarios where victory is impossible, rendering the curriculum meaningless. The acceptance threshold maintains consistency with target performance standards. If needed, it could also be designed autonomously by the Designer. This demonstrates that our framework balances automation with controllability while maintaining strong generalization across diverse scenarios.
>
> Second, to be honest, we have not configured numerous rules for EvoCurr to ensure the final results. On the contrary, the rules we use only include some basic rule introductions and constraints for the LLM (such as not using non-existent functions, etc.). Based on our architectural design, we stimulate the LLM's potential to achieve good performance across multiple scenarios while maintaining generalizability. In our tests, removing any existing rule significantly causes the LLM to generate erroneous content or fail to return content in the required format. Therefore, it is difficult to analyze the respective contribution of each rule to EvoCurr's performance improvement. We sincerely thank you for the suggestion of adding ablation experiments. We are also considering adding new content to enhance EvoCurr's performance and implement the design of ablation experiments.
>
> Third, We do not have a clear understanding of what "properties" refers to in your comment. If you could provide clarification on this, we would greatly appreciate it. And regarding EvoCurr's generalizability, for the Designer, it only requires the experimental scenario and task requirements to be specified, and the Designer can automatically generate curriculum designs. For the Solver, the two scenarios of SC2 and Overcooked have demonstrated that in scenarios where RL can be applied and implementations can be achieved through code, EvoCurr can adapt well with simple modifications and achieve strong performance, demonstrating robust generalization capabilities.
>
> We would like to once again express our sincere gratitude for your thoughtful review and valuable feedback. Your constructive comments have been instrumental in helping us improve the quality and clarity of our work. We hope that our responses and revisions have adequately addressed your concerns. Should you have any further questions or require additional clarification on any points, we would be more than happy to provide further discussion. Thank you for your time and consideration.

---

### Official Review · Reviewer_kEh9 · 2025-11-01

**Soundness:** 2
**Presentation:** 3
**Contribution:** 1
**Rating:** 2
**Confidence:** 4

**Summary:**

This paper proposes EvoCurr, an adversarial-like curriculum learning framework. It has two cooperating agents: a Designer, which creates tasks with gradually increasing difficulty, and a Solver, which solves them. The goal is to help large language models (LLMs) handle more complex decision-making tasks. The authors tested their approach in two environments: StarCraft II micro-management and Overcooked, and claim their method achieves improved performance compared to direct baselines.

**Strengths:**

- The paper structure and presentation are clear and easy to follow.
- Figures and tables are well-made, helping readers easily grasp the main idea.

**Weaknesses:**

- The proposed approach isn't really new; similar frameworks have already been explored before. Even though the authors applied it to LLMs, it doesn't really introduce anything new.
- The method relies too heavily on heuristic choices and manually set hyper-parameters (like the difficulty measure $d(C)$ and acceptance threshold $\tau$). This makes the method seem overly simplistic.
- The experimental environments feel somewhat toy-like, raising questions about the generalizability of the results.
- Using acceptance rate as the curriculum metric seems arbitrary. It might be more reasonable to use loss signals or something more meaningful.
- The contributions listed in the intro feel exaggerated since the core ideas have appeared previously.

**Questions:**

- Can the authors provide more discussion about potential generalization beyond these simplified scenarios?

---

> ### Author Response · Authors · 2025-11-24
> **Answers for questions part1**
>
> Thank you sincerely for your thorough review and thoughtful questions. I truly appreciate the time and care you've invested in evaluating our work. Please allow me to address your concerns and clarify some aspects that may not have been adequately explained in the manuscript.
>
> **Regarding the novelty of curriculum learning methods**: While I acknowledge that curriculum learning itself is not a novel concept, I'd like to clarify that our contribution lies elsewhere. In most existing applications, curricula are manually designed to ensure steady performance improvement for LLMs or agents. Our framework, however, enables the Designer to automatically construct and dynamically adjust curricula based on the Solver's real-time performance. This automation significantly reduces the complexity inherent in manual curriculum design while enhancing generalization across diverse scenarios. The Designer adaptively modulates task difficulty throughout the learning process, ensuring rational task construction and enabling the Solver to achieve robust performance across varying difficulty levels and contexts. This adaptive, automated approach is what we believe advances the field.
>
> **Regarding the difficulty measure d(C) and acceptance threshold**: I understand your concern about the framework's reliance on heuristic choices and manually set hyperparameters. I'd like to clarify our design philosophy and address this concern directly.
> First, ensuring that the difficulty measure d(C) monotonically increases is indeed necessary to prevent performance regression—this is a fundamental and well-established principle in any curriculum learning algorithm or application. However, to preserve our framework's generalization capability, we intentionally avoided scenario-specific manual configurations for d(C). Instead, d(C) is primarily determined autonomously by the LLM itself. This design choice ensures that the framework does not become overly dependent on manually crafted difficulty metrics, thereby reducing rather than increasing reliance on manual heuristics.
>
> **Regarding the acceptance threshold**: it serves specifically to evaluate whether the Solver can adequately complete the current task. We configured it manually to maintain consistency with our target performance standards for the Solver. However, I want to emphasize that this represents a deliberate design choice rather than a limitation. If the sole objective were to ensure continuous performance improvement without specific performance targets, the acceptance threshold could equally be designed autonomously by the Designer rather than relying on manual specification. This flexibility demonstrates that our framework balances automation with controllability, allowing practitioners to inject domain knowledge when needed while maintaining generalization across scenarios.
>
> **Regarding the experimental environments**: We respectfully disagree that our experimental environments are "toy-like." Our evaluation is grounded in **StarCraft II**, which is established by Vinyals et al. (Grandmaster level in StarCraft II using multi-agent reinforcement learning, Nature 2019) as one of the most challenging and strategically complex environments for AI, featuring a combinatorial action space of approximately **$10^{26}$ per step**. Our setup specifically targets the computationally demanding subset of this domain—large-scale micro-management—which serves as a rigorous stress test for decision-making.
> 1. **StarCraft II Complexity (Combat Scale)**: Unlike standard benchmarks (e.g., SMAC, SMACV2) that focus on small-scale skirmishes, our scenarios simulate "super late-game" combat. Agents must control over 50 heterogeneous units simultaneously, managing diverse active abilities (e.g., Psi Storm, Siege Mode, Stimpack). This complexity significantly exceeds that of default AI opponents, as our adversaries utilize LLM-generated behavior trees capable of sophisticated tactical reasoning.
>
> 2. **Overcooked (Coordination Standard)**: We also include Overcooked, a canonical benchmark for multi-agent coordination, to ensure our framework is evaluated across both high-dimensional competitive control and precise cooperative tasks.
> By conquering scenarios that operate within such a vast action space and solving established coordination benchmarks, EvoCurr demonstrates robustness and generalizability well beyond toy problems.
>
> By conquering scenarios that operate within such a vast action space and solving established coordination benchmarks, EvoCurr demonstrates robustness and generalizability well beyond toy problems.

---

> ### Author Response · Authors · 2025-11-24
> **Answers for questions part2**
>
> **Regarding the acceptance rate calculation**: In EvoCurr, acceptance rates are derived directly from empirical testing in the actual experimental environments—battle win rates in StarCraft II (SC2) or task completion rates in Overcooked. I believe this approach provides an intuitive and direct reflection of the Solver's capability on the current task. Rather than being arbitrary, this method grounds our evaluation in concrete, measurable outcomes.
>
> The use of acceptance rates as a metric is further supported by research in the field. For instance, in  *Efficient Reinforcement Learning with Milestone Aggregation in Asynchronous Distributed Training for RTS Games*(https://www.sciencedirect.com/science/article/abs/pii/S0925231225011026) and *in Efficient Reinforcement Learning for Full-length Game of StarCraft II* (https://arxiv.org/abs/2209.11553), both studies advocate for metrics that are directly tied to performance in real-world settings, underscoring the validity of using such empirical results as a gauge of an agent's success in the task at hand.
>
> **Regarding the potential generalization for EvoCurr**: The core of EvoCurr actually lies in the automated design of curricula. In traditional curriculum learning methods, there are essentially no generalizable curriculum learning approaches - they all require specific curriculum designs for specific scenarios. EvoCurr not only enables automated curriculum construction given a final task, greatly reducing the difficulty of building curricula, but also integrates with the Solver to verify curriculum effectiveness and automatically adjust curriculum rationality. Meanwhile, EvoCurr also possesses generalizability. For experimental scenarios like SC2 and Overcooked, it only requires modifying the experimental environment and final task description to complete the construction of curriculum learning, giving curriculum learning methods good generalizability as well.
>
> Once again, I want to express my deep appreciation for your careful, responsible review and valuable feedback. Your questions have helped me identify areas where our presentation could be clearer, and I hope my responses have addressed your concerns. I remain open to further discussion and genuinely grateful for your review on our work.

---

> > ### Comment · Reviewer_kEh9 · 2025-11-25
> >
> > Thanks to the authors for their detailed response, which clarified my misunderstanding regarding the complexity of the experimental environments. I acknowledge that the StarCraft II and Overcooked scenarios indeed represent challenging and non-toy benchmarks.
> >
> > However, my primary concern remains the heuristic nature of the proposed approach. While automating a curriculum progression from easy to difficult tasks is valuable, this incremental design itself is not particularly surprising or novel. Although the authors clearly invested substantial engineering effort, it is not clear to me that this contribution will significantly impact or inspire future research directions.
> >
> > Taking all points into consideration, I slightly increase my rating to 4 (borderline), reflecting a more balanced assessment.

---

> > > ### Author Response · Authors · 2025-11-25
> > > **Genuine Appreciation for Your Feedback and Sincere Expression of Our Perspective**
> > >
> > > Thank you for your thoughtful follow-up and for taking the time to reconsider your evaluation. We sincerely appreciate your acknowledgment of the complexity of the StarCraft II and Overcooked environments, as well as your careful reflection on our responses. And we are grateful for your updated assessment. However, we believe our method offers genuine value, as it provides a simple yet effective pathway toward reducing human effort and enabling more scalable curriculum design.
> > >
> > > Our motivation for proposing this method stems from the fact that existing curriculum-learning approaches and most reinforcement learning methods rely heavily on human experts to meticulously design the environments. In contrast, our method requires only the initial and final environments provided by experts, and can automatically generate the intermediate environments that would otherwise demand substantial time and effort to create. This progressive, staged design is— in our view— the simplest yet highly effective approach.
> > >
> > > Moreover, in our method, the intermediate environments are automatically generated in large quantities, rather than following a single linear progression. When the task difficulty becomes unreasonable, the system also dynamically adjusts by slightly reducing the difficulty.
> > >
> > > We believe that automated environment generation holds tremendous potential on the path toward general artificial intelligence (AGI). It will significantly accelerate future research in reinforcement learning and LLM-based agents, enabling more diverse and higher-quality evaluation benchmarks and ultimately allowing agents to achieve stronger performance and continual improvement.
> > >
> > > Like the 23 maps in SMAC have been studied for many years, yet innovation of maps has been scarce. The fundamental reason lies in the difficulty of estimating model capabilities and judging the reasonableness of task difficulty. However, using our method, we can explore the limits of what models can achieve to a certain extent, and have a clearer and more rational approach to assessing task difficulty. This can help us rapidly design more high-quality maps and, based on this, explore where the true capability limits of agents lie, enabling us to better improve agent capabilities and performance.
> > >
> > > We hope this clarification further highlights the significance and potential impact of our work. We sincerely thank you for your careful review and valuable feedback.

---

### Official Review · Reviewer_YKRz · 2025-11-04

**Soundness:** 3
**Presentation:** 3
**Contribution:** 3
**Rating:** 6
**Confidence:** 3

**Summary:**

The paper introduces EvoCurr, a self-evolving curriculum framework for inference-time problem solving with LLMs. A Designer LLM proposes the next task configuration; a Solver either (i) generates executable behavior-tree code (open-loop, interpretable) or (ii) trains a MARL policy (closed-loop) under a fixed budget. Two rules named accepted-floor (no regression below the last mastered difficulty) and a feasibility gate (syntax/logic/runtime checks) yield monotonic skill acquisition without manual difficulty metrics. The method reaches 90% win-rate on 12 SC2 micro-management tasks after 4–6 curriculum steps and delivers ~20% higher order completion on Overcooked than direct training under matched budgets.

**Strengths:**

- Inference-time curriculum with simple rules (accepted-floor + feasibility gate) removes the need for hand-crafted difficulty metrics and schedules.

- The behavior trees and the syntax & code critic make debugging and analysis practical.

- Compelling results on 12 SC2 tasks and Overcooked with clear acceptance criteria and curriculum traces.

**Weaknesses:**

- The major concern is the limited baseline. To more clearly clarify the advantage of the proposed method, the authors are recommended to compare the proposed method to stronger curriculum RL baselines  or strong search-based planners beyond the “one-shot direct code” baseline.

- Directly using code as policy may limit the capacity. Behavior-tree size and LLM context length may cap complexity; ablations on tree depth/lines of code vs performance would be useful.

**Questions:**

- Could the authors report the env-steps, compiles, wall-clock per accepted curriculum vs direct baselines to demonstrate its converging process?

---

> ### Author Response · Authors · 2025-11-24
> **Answers for questions part1**
>
> We sincerely thank you for your thoughtful and constructive feedback. We deeply appreciate the time and effort invested in evaluating our work, and we are grateful for the valuable insights and suggestions provided. We have carefully considered your comments and will clarify the questions on the following contents. We hope that our responses adequately address the concerns and help improve the clarity and quality of the paper.
> First, we compared the performance of the strong search-based planner ( like Claude code, Codex and Cursor )  and more baselines, and implemented EvoCurr on GPT-5 and Claude-4. Below is a comparison table of the results for different models:
> ## Score Table
>
> | Task | EvoCurr (DeepSeek v3.1) | EvoCurr (Claude-4) | EvoCurr (GPT-5) | DeepSeek | GPT-5 | Claude-4 | Gemini-2.5 | Claude_code | Codex | Cursor |
> | :--- | :---: | :---: | :---: | :---: | :---: | :---: | :---: | :---: | :---: | :---: |
> | bush (TVP) | **0.699841** | 0.636 | 0.696 | 0.338 | 0.436 | 0.357 | 0.499 | 0.194 | 0.279 | 0 |
> | corridor (TVP) | **0.714642** | 0.610 | 0.620 | 0.561 | 0.516 | 0.638 | 0.264 | 0.192 | 0.213 | 0 |
> | corner (TVP) | 0.551 | 0.637 | 0.659 | **0.666** | 0.526 | 0.625 | 0.559 | 0.282 | 0.275 | 0 |
> | flat (TVP) | **0.699631** | 0.607 | 0.629 | 0.412 | 0.567 | 0.390 | 0.308 | 0.216 | 0.221 | 0 |
> | main (TVP) | 0.728 | 0.657 | 0.667 | 0.697 | 0.632 | 0.677 | **0.774** | 0.343 | 0.408 | 0 |
> | ramp (TVP) | 0.723 | 0.635 | 0.681 | 0.457 | 0.484 | **0.745** | 0.662 | 0.218 | 0.239 | 0 |
> | bush (TVZ) | **0.732238** | 0.670 | 0.647 | 0.612 | 0.342 | 0.647 | 0.368 | 0.465 | 0.318 | 0 |
> | corridor (TVZ) | **0.697976** | 0.650 | 0.575 | 0.588 | 0.569 | 0.656 | 0.598 | 0.343 | 0.313 | 0 |
> | corner (TVZ) | **0.840462** | 0.742 | 0.621 | 0.415 | 0.698 | 0.776 | 0.801 | 0.741 | 0.621 | 0 |
> | flat (TVZ) | **0.626284** | 0.612 | 0.531 | 0.361 | 0.297 | 0.493 | 0.377 | 0.311 | 0.278 | 0 |
> | main (TVZ) | 0.702 | **0.760** | 0.753 | 0.685 | 0.433 | 0.431 | 0.668 | 0.455 | 0.451 | 0 |
> | ramp (TVZ) | **0.746564** | 0.652 | 0.706 | 0.450 | 0.454 | 0.682 | 0.294 | 0.454 | 0.293 | 0 |
>
> ---
>
> ## Winning Rate Table (%)
>
> | Task | EvoCurr (DeepSeek v3.1) | EvoCurr (Claude-4) | EvoCurr (GPT-5) | DeepSeek | GPT-5 | Claude-4 | Gemini-2.5 | Claude_code | Codex |  Cursor |
> | :--- | :---: | :---: | :---: | :---: | :---: | :---: | :---: | :---: | :---: | :---: |
> | bush (TVP) | **100** | **100** | **100** | 0 | 0 (equal 3 times) | 10 | 0 (equal 4 times) | 0 | 0 | 0 |
> | corridor (TVP) | **100** | 70 (equal 3 times) | 90 (equal 1 times) | 80 | 80 (equal 2 times) | **100** | 0 (equal 2 times) | 0 | 0 | 0 |
> | corner (TVP) | 90 | **100** | **100** | **100** | 90 | **100** | 30 | 0 | 0 | 0 |
> | flat (TVP) | **100** | **100** | **100** | 0 | **100** | 40 | 0 | 0 | 0 | 0 |
> | main (TVP) | **100** | **100** | **100** | **100** | **100** | **100** | **100** | 0 (equal 2 times) | 0 | 0 |
> | ramp (TVP) | **100** | **100** | **100** | 10 | 0 (equal 4 times) | **100** | **100** | 0 | 0 | 0 |
> | bush (TVZ) | **100** | 90 (equal 1 times) | **100** | **100** | 0 | **100** | 0 | 40 (equal 1 times) | 0 (equal 5 times) | 0 |
> | corridor (TVZ) | **100** | **100** | 90 | 90 | 0 (equal 4 times) | **100** | **100** | 0 (equal 2 times) | 0 (equal 4 times) | 0 |
> | corner (TVZ) | **100** | **100** | **100** | 30 | **100** | **100** | **100** | **100** | **100** | 0 |
> | flat (TVZ) | **100** | **100** | 90 | 0 | 0 | 0 | 0 | 0 | 0 | 0 |
> | main (TVZ) | **100** | **100** | **100** | **100** | 0 | 0 | **100** | 10 (equal 2 times) | 20 (equal 2 times) | 0 |
> | ramp (TVZ) | **100** | **100** | **100** | 50 | 60 (equal 1 times) | **100** | 0 | 20 (equal 5 times) | 0 | 0 |
>
> ---
>
> **Additionally:** We tested Cursor with Grok4, but it consistently generated incorrect attribute or function names.
>
> The data in these tables clearly demonstrate that the implementation of EvoCurr outperforms the search-based planner (the only advantage of the search-based planner is that when combined with models with strong code capabilities, the probability of encountering incorrect function or attribute names is lower). It also proves the generalization ability of the EvoCurr model, as it can be implemented across multiple models.

---

> ### Author Response · Authors · 2025-11-24
> **Answers for questions part2**
>
> Second, the impact of Behavior-tree size and LLM context length on the experimental results is minimal. Comparing EvoCurr implemented with Deepseek, Claude, and GPT, it is evident that the context length or Behavior-tree size of the final generated result using Claude is significantly larger than that of the other models. However, the final implementation results do not show significant differences (and in some cases, a larger context length or Behavior-tree size actually leads to worse performance)
>
> Third, to be honest, we were not able to fully understand the meaning of env-steps, compiles, and wall-clock per accepted curriculum. Therefore, based on our understanding, we provided the curriculum environment and accepted time for the bush (PVT) and bush (ZVT) scenarios during the EvoCurr process. If our understanding is incorrect, we will modify the data according to your suggestions. If there is no issue, we would appreciate it if you could clarify whether the curriculum environment and accepted time for all scenarios are still required, and we will add them accordingly.
> ## Bush ( TVP )  Processing Table:
>
> | Task | Accepted Time | Agents Composition | Agents Abilities | Enemies Composition | Enemies Abilities |
> | :--- | :---: | :--- | :--- | :--- | :--- |
> | **task1** | 0:06:17 | Marine x12, Marauder x6, Medivac x2, SiegeTank x4 | Stimpack, Heal, SiegeMode | Zealot x5, Stalker x8, HighTemplar x3, Sentry x4 | Charge, BlinkTech, PsiStormTech, ForceField |
> | **task2** | 0:02:47 | Marine x16, Marauder x8, Medivac x3, SiegeTank x5, Ghost x4, VikingFighter x4 | Stimpack, Heal, SiegeMode, PersonalCloaking | Zealot x8, Stalker x10, HighTemplar x5, Sentry x6, Colossus x2, Disruptor x2 | Charge, BlinkTech, PsiStormTech, ForceField, ExtendedThermalLance |
> | **task3** | 0:02:15 | Marine x18, Marauder x10, Medivac x3, SiegeTank x6, Ghost x6, VikingFighter x6, Cyclone x4, WidowMine x4 | Stimpack, Heal, SiegeMode, PersonalCloaking, Burrow | Zealot x12, Stalker x12, HighTemplar x6, Sentry x8, Colossus x3, Disruptor x3, Tempest x2 | Charge, BlinkTech, PsiStormTech, ForceField, ExtendedThermalLance |
> | **task4** | 0:17:11 | Marine x20, Marauder x12, Medivac x4, SiegeTank x6, Ghost x8, VikingFighter x8, Cyclone x6, WidowMine x6, Raven x2 | Stimpack, Heal, SiegeMode, PersonalCloaking, Burrow, HunterSeeker | Zealot x15, Stalker x14, HighTemplar x8, Sentry x10, Colossus x4, Disruptor x4, Tempest x3, Carrier x2 | Charge, BlinkTech, PsiStormTech, ForceField, ExtendedThermalLance |
> | **task5** | 0:02:39 | Marine x20, Marauder x12, Medivac x4, SiegeTank x6, Ghost x8, VikingFighter x8, Cyclone x8, WidowMine x8, Raven x3, Liberator x4 | Stimpack, Heal, SiegeMode, PersonalCloaking, Burrow, HunterSeeker | Zealot x15, Stalker x15, HighTemplar x9, Sentry x10, Colossus x4, Disruptor x4, Tempest x4, Carrier x4 | Charge, BlinkTech, PsiStormTech, ForceField, ExtendedThermalLance |
> | **task6** | 0:05:58 | Marine x20, Marauder x12, Medivac x4, SiegeTank x6, Ghost x8, VikingFighter x8, Cyclone x8, WidowMine x8, Raven x3, Liberator x4 | Stimpack, Heal, SiegeMode, PersonalCloaking, Burrow, HunterSeeker | Zealot x15, Stalker x15, HighTemplar x9, Sentry x10, Colossus x4, Disruptor x4, Tempest x5, Carrier x4 | Charge, BlinkTech, PsiStormTech, ForceField, ExtendedThermalLance |
>
> ##  Bush (TvZ)  Processing Table:
>
> | Task | Accepted Time | Agents Composition | Agents Abilities | Enemies Composition | Enemies Abilities |
> | :--- | :---: | :--- | :--- | :--- | :--- |
> | **task1** | 0:07:15 | Marine x12, Marauder x6, Medivac x2, SiegeTank x4 | Stimpack, Heal, SiegeTech | Zergling x20, Roach x8, Hydralisk x6 | zerglingmovementspeed, GlialReconstitution, hydraliskspeed |
> | **task2** | 0:14:40 | Marine x16, Marauder x8, Medivac x3, SiegeTank x5, Ghost x4, VikingFighter x4 | Stimpack, Heal, SiegeTech, PersonalCloaking | Zergling x30, Roach x12, Hydralisk x8, Baneling x12, Corruptor x5 | zerglingmovementspeed, GlialReconstitution, hydraliskspeed, CentrificalHooks |
> | **task3** | 1:01:39 | Marine x18, Marauder x10, Medivac x4, SiegeTank x6, Ghost x6, VikingFighter x6, Cyclone x5, WidowMine x5, Raven x2 | Stimpack, Heal, SiegeTech, PersonalCloaking, Burrow, HunterSeeker | Zergling x40, Roach x15, Hydralisk x10, Baneling x18, Corruptor x8, Lurker x4, Infestor x2, Viper x2, Overseer x2, Queen x3, BroodLord x2 | zerglingmovementspeed, GlialReconstitution, hydraliskspeed, CentrificalHooks, Burrow, InfestorPeristalsis, Abduct, Detection, Transfusion |
> | **task4** | 2:52:56 | Marine x20, Marauder x12, Medivac x4, SiegeTank x6, Ghost x8, VikingFighter x8, Cyclone x8, WidowMine x8, Raven x3, Liberator x4 | Stimpack, Heal, SiegeTech, PersonalCloaking, Burrow, HunterSeeker | Zergling x60, Baneling x24, Roach x15, Hydralisk x10, Lurker x6, Corruptor x10, Infestor x3, Viper x4, Overseer x3, Queen x4, BroodLord x4 | zerglingmovementspeed, CentrificalHooks, GlialReconstitution, hydraliskspeed, Burrow, InfestorPeristalsis, Abduct, Detection, Transfusion |

---

> ### Author Response · Authors · 2025-11-24
> **Answers for questions part3**
>
> Finally, we would like to once again express our sincere gratitude to the reviewers for their insightful comments and constructive feedback. Your thoughtful suggestions have significantly improved the quality of our work, and we truly appreciate the time and effort you have dedicated to reviewing our manuscript. We hope that our revisions adequately address your concerns and enhance the clarity of our contributions.

---

### Comment · Area_Chair_ViZr · 2025-11-25

Dear Reviewer, thank you for reviewing for ICLR. Since the discussion deadline is coming soon, could you please take a look at the author's rebuttal, respond to their comments, and update your rating as well? Thanks!

---

### Note · Authors · 2025-12-01

I have read and agree with the venue's withdrawal policy on behalf of myself and my co-authors.